# An innate contribution of human nicotinic receptor polymorphisms to COPD-like lesions

Julie Routhier[1], Stéphanie Pons[2], Mohamed Lamine Freidja [1,8], Véronique Dalstein[1,3], Jérôme Cutrona[1], Antoine Jonquet[1], Nathalie Lalun[1], Jean-Claude Mérol[1,4], Mark Lathrop[5], Jerry A. Stitzel[6], Gwenola Kervoaze[7], Muriel Pichavant[7], Philippe Gosset[7], Jean-Marie Tournier[1], Philippe Birembaut[1,3,9], Valérian Dormoy [1,9✉] & Uwe Maskos [2,9✉]

Chronic Obstructive Pulmonary Disease is a generally smoking-linked major cause of morbidity and mortality. Genome-wide Association Studies identified a locus including a non-synonymous single nucleotide polymorphism in *CHRNA5*, rs16969968, encoding the nicotinic acetylcholine receptor α5 subunit, predisposing to both smoking and Chronic Obstructive Pulmonary Disease. Here we report that nasal polyps from rs16969968 non-smoking carriers exhibit airway epithelium remodeling and inflammation. These hallmarks of Chronic Obstructive Pulmonary Disease occur spontaneously in mice expressing human rs16969968. They are significantly amplified after exposure to porcine pancreatic elastase, an emphysema model, and to oxidative stress with a polymorphism-dependent alteration of lung function. Targeted rs16969968 expression in epithelial cells leads to airway remodeling in vivo, increased proliferation and production of pro-inflammatory cytokines through decreased calcium entry and increased adenylyl-cyclase activity. We show that rs16969968 directly contributes to Chronic Obstructive Pulmonary Disease-like lesions, sensitizing the lung to the action of oxidative stress and injury, and represents a therapeutic target.

[1] Université de Reims Champagne-Ardenne, Inserm, P3Cell UMR-S1250 Reims, France. [2] Institut Pasteur, Université de Paris, Integrative Neurobiology of Cholinergic Systems, CNRS UMR 3571 Paris, France. [3] Department of Biopathology, CHU of Reims, Reims, France. [4] Department of Otorhinolaryngology, CHU of Reims, Reims, France. [5] McGill University Genome Center, Montréal, QC, Canada. [6] Institute for Behavioral Genetics, University of Colorado, Boulder, CO, USA. [7] University of Lille, CNRS UMR9017, Inserm U1019, CHU Lille, Institut Pasteur de Lille, CIIL - Center for Infection and Immunity of Lille, Lille, France. [8] Present address: Department of Biochemistry and Microbiology, Faculty of Sciences, University of M'sila, M'sila, Algeria. [9] These authors contributed equally: Philippe Birembaut, Valérian Dormoy, Uwe Maskos. ✉email: valerian.dormoy@univ-reims.fr; umaskos@pasteur.fr

Chronic obstructive pulmonary disease (COPD) is a major health problem and expected to shortly become the third leading cause of worldwide mortality and the fifth leading cause of morbidity[1–3]. Independent and robust genome-wide association studies (GWAS)[4,5], and meta-analyses[6,7], have linked this disease to a locus on chromosome 15q25. There is one signal at the *IREB2* gene, also part of this locus. This was explored in a mouse model of cigarette smoke–induced bronchitis and emphysema[8]. 15q25 also encompasses three genes, *CHRNA3*, *CHRNA5*, and *CHRNB4*, coding for the α3, α5, and β4 subunits of the nicotinic acetylcholine receptor (nAChR). This locus has been previously linked to tobacco smoking and nicotine consumption, using GWAS, validated in our transgenic mouse and rat models for increased nicotine intake and relapse[9–11], and summarized in our recent review[12]. However, it is unclear whether the associations between SNPs and COPD are solely due to smoking. It is also important to note that a substantial percentage of COPD patients are exposed to biomass fuel-derived smoke, and to unidentified causes[13,14]. Moreover, the non-synonymous SNP rs16969968 (α5SNP) has been linked to lung cancer[15,16], a major comorbidity associated with COPD[12,17].

Importantly, two GWAS identified a potential innate, smoking-independent component of the disease, in carriers of the nicotinic receptor SNPs[4,18]. Specifically, at equal smoking levels, subjects carrying the SNP were significantly more prone to develop the disease. Wilk et al.[18] concluded that there is an important role for the CHRNA5/3 region as a genetic risk factor for airflow obstruction that may be independent of smoking. Siedlinski et al.[19] carried out mediation analysis to identify direct and indirect effects on COPD risk. Their results with 3424 COPD cases and 1872 unaffected controls revealed that effects of two linked variants (rs1051730 and rs8034191) in the AGPHD1/CHRNA3 cluster on COPD development are significantly, yet not entirely, mediated by the smoking-related phenotypes. Approximately 30% of the total effect of variants in the AGPHD1/CHRNA3 cluster on COPD development was mediated by pack years. Simultaneous analysis of modestly ($r^2 = 0.21$) linked markers in *CHRNA3* and *IREB2* revealed that an even larger (~42%) proportion of the total effect of the *CHRNA3* locus on COPD was mediated by pack years after adjustment for an *IREB2* single nucleotide polymorphism. This would suggest that potentially 60% of the effect is not linked to smoking. Moreover, rs16969968 was found to independently confer risk of lung cancer, COPD and smoking intensity in 9270 non-hispanic white subjects from the National Lung Screening Trial[20], re-iterating the concept of a triple whammy. On the other hand, two studies using the UK Biobank and additional COPD cohorts[7,21] carrying out GWAS for impaired lung function, and COPD on the basis of prebronchodilator spirometry, respectively, did not identify an association in non-smokers, illustrating that there is conflicting evidence in non-smokers.

COPD is characterized by repeated lesions of the airways consecutive to local aggressions such as chronic cigarette smoke or biomass fuel-derived microparticles. A deregulated abnormal wound repair that involves proliferation, migration and, differentiation of progenitor cells[22] is responsible for epithelial remodeling with hyperplasia of basal epithelial cells, squamous metaplasia and hyperplasia of goblet cells. These lesions progressively induce bronchiolar and bronchial obstruction, favor infections, and lead to emphysema and respiratory failure. We have previously reported the role of the α5 subunit in migration of normal airway epithelium cells, and in tumor invasion in lung cancers[23,24]. This subunit is expressed by basal cells in the epithelium[23,25], which are considered to act as progenitor cells[26,27]. In addition, we reported the expression and the localization of the α5 subunit in airway epithelial cells from bronchi and bronchioles in non-smokers[28].

In this work, we study the role of α5SNP in the remodeling of the airways and inflammation, independently of its established role in nicotine dependence. We have used mouse models, particularly humanized mice expressing α5SNP[29], to determine its role in the development of epithelial remodeling and inflammation without any experimental manipulation, and to established nicotine-independent murine models mimicking these alterations, after intra-tracheal porcine pancreatic elastase (PPE) treatment[30] and oxidative stress. Oxidative stress-related exposure to environmental factors as well as protease-antiprotease imbalance are essential for the development of COPD[31,32]. To further elucidate the underlying mechanisms, we have analyzed the impact of α5SNP expression in murine and human epithelial tissues in vivo and in vitro (Fig. 1a). We identify a crucial role of α5SNP in the smoking-independent contribution to COPD-like lesions.

## Results

**α5SNP is spontaneously associated with airway epithelial remodeling and emphysema.** First, we analyzed samples of nasal polyps from 123 non-smoking patients with the three CHRNA5 genotypes at position 78590583 on GRCh38: wild-type (WT, G/G), heterozygous (HT, G/A), and homozygous (HO, A/A) for rs16969968 (α5SNP). As nasal polyps are lined by a pseudostratified airway epithelium, similar to the bronchial epithelium, they represent a relevant model for human epithelial cell physiopathology, which has been used and validated by us and others[33–38]. In a multivariate analysis, the presence of the α5SNP in HT or HO α5SNP patients compared with patients carrying the WT allele, was significantly associated with inflammation of the lamina propria, goblet cell hyperplasia, and global epithelial remodeling (Fig. 1, Table 1, and Table S1).

To decipher the functional link between α5SNP and pulmonary homeostasis, we took advantage of a transgenic mouse model knock-in (KI) for α5SNP[29], whose intra-pulmonary airway epithelial cells express the nAChR α5 subunit in its mutated form (Fig. S1). The α5SNP mice spontaneously exhibited remodeled airway epithelium with hyperplasia of goblet cells in the nasal respiratory epithelium as well as in the extra-lobular bronchi (Fig. 2a, Fig. S2). This increase in goblet mucus-producing cells[39], which are Alcian blue-positive, was accompanied by a decrease of Club cells, CC10 positive, known for their anti-inflammatory properties[40], both in proximal and distal airways (Fig. 2a and b). There were no modifications in the expression of Ki-67 (Fig. S3). No change was observed in the population of ciliated cells, β-tubulin positive, in distal airways. Importantly, emphysema naturally occurred at 54 weeks in α5SNP HO mice, whereas a significant increase in inflammatory cells, especially macrophages, was already present at 24 weeks (Fig. 2c and d, Fig. S4). We confirmed the inflammatory component associated with our transgenic mouse model by an instillation of polidocanol[41,42] into the nostrils to induce local airway lesions followed by wound repair and inflammation, a condition similar to the initial lesions observed in COPD. The recruitment of inflammatory cells was increased in α5SNP mice as shown for eosinophils, neutrophils, and macrophages, and this was associated with the mobilization of neutrophils, absent in WT mice (Fig. S5). Moreover, we confirmed the emphysema component associated with our transgenic mouse model with a rapid induction of alveolar destruction by an intra-tracheal instillation of porcine pancreatic elastase (PPE)[43–45]. α5SNP mice presented increased emphysema when compared to WT mice (Fig. S6a).

We did not observe an increased inflammatory cell recruitment nor higher cytokine concentrations in PPE-treated WT mice, and no modifications in α5SNP mice except an increased level of

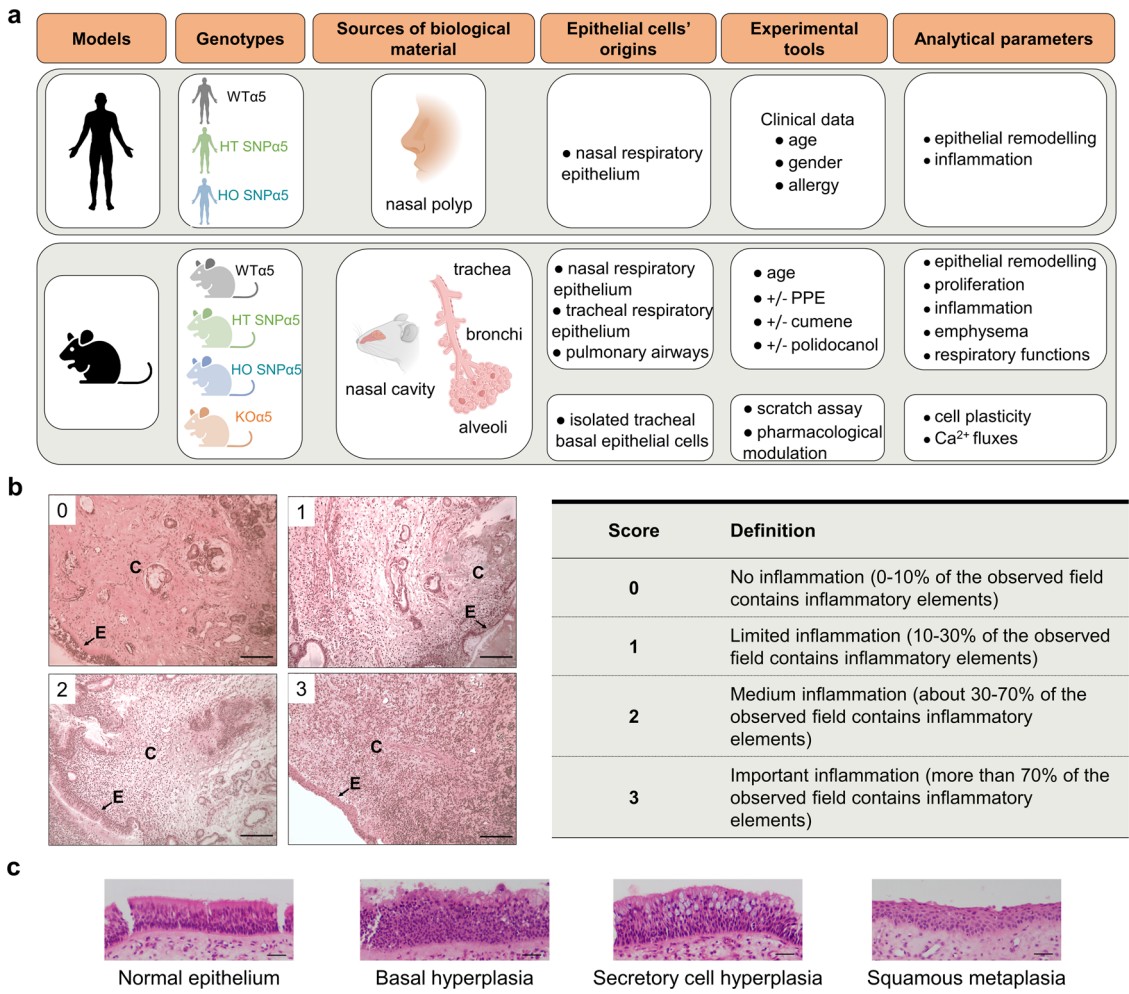

**Fig. 1 α5SNP expression and COPD-like remodeling in human airway epithelium. a** Scheme highlighting the experimental systems (created with BioRender.com). Inflammation of the *lamina propria* (**b**), and epithelial phenotypes (**c**) were quantified from 123 human nasal polyp samples on three micrographs by two histo-pathologists (bar = 100 μm, E = epithelium, C = chorion). Effect of genotype, age, gender and, allergy/asthma on human epithelial phenotype and inflammation was analyzed in multivariate regression analysis (Table 1).

**Table 1 Multivariate regression analysis of the effect of patients' genotype and age/gender/allergy-asthma on epithelium remodeling.**

| Variable | Inflammation | | Secretory cell hyperplasia | | Normal epithelium | |
|---|---|---|---|---|---|---|
| | Odds-ratio [CI 95 %] | *p* | Odds-ratio [CI 95 %] | *p* | Odds-ratio [CI 95 %] | *p* |
| Genotype | | | | | | |
| WT (G/G) | 1 | – | 1 | – | 1 | – |
| HT (G/A) | 3.5 [1.39–8.79] | 0.008 | 5.18 [2–13.39] | 0.001 | 0.43 [0.17–1.07] | 0.068 |
| HO (A/A) | 6.47 [1.92–21.83] | 0.003 | 9.62 [2.39–38.63] | 0.001 | 0.2 [0.53–0.74] | 0.016 |
| Age | 0.97 [0.94–0.99] | 0.015 | 0.99 [0.96–1.02] | 0.429 | 1.01 [0.98–1.05] | 0.505 |
| Gender | | | | | | |
| M | 1 | - | 1 | - | 1 | - |
| F | 1.69 [0.74–3.86] | 0.212 | 0.9 [0.38–2.11] | 0.81 | 0.98 [0.42–2.29] | 0.962 |
| Allergy/Asthma | | | | | | |
| No | 1 | - | 1 | - | 1 | - |
| Yes | 0.63 [0.28–1.42] | 0.266 | 0.94 [0.39–2.99] | 0.899 | 1.64 [0.7–3.85] | 0.255 |

*CI* confidence interval, *WT (G/G)* wild-type, *HT (G/A)* heterozygous for α5 single nucleotide polymorphism, *HO (A/A)* homozygous for α5 single nucleotide polymorphism.
Effect of genotype, age, gender and allergy/asthma on human epithelial phenotype and inflammation was analyzed in multivariate regression analysis. Two epithelial phenotypes (secretory cell hyperplasia and normal epithelium) and sub-mucosa inflammation were quantified. Effect of patients' genotype and age/gender/allergy-asthma on epithelium remodeling was assessed by a multivariate regression analysis with presentation of odds ratios.

CCL20 (Fig. S6b). But importantly, both epithelial remodeling and emphysema were recapitulated. These observations correspond to a spontaneous mild lung phenotype resembling COPD in mice bearing the α5SNP.

**Oxidative stress induces more severe lung inflammation and remodeling in α5SNP mice.** COPD lesions are mostly related to chronic exposure to oxidative pollutants and are associated with an imbalance between oxidants and antioxidants[46]. To emphasize

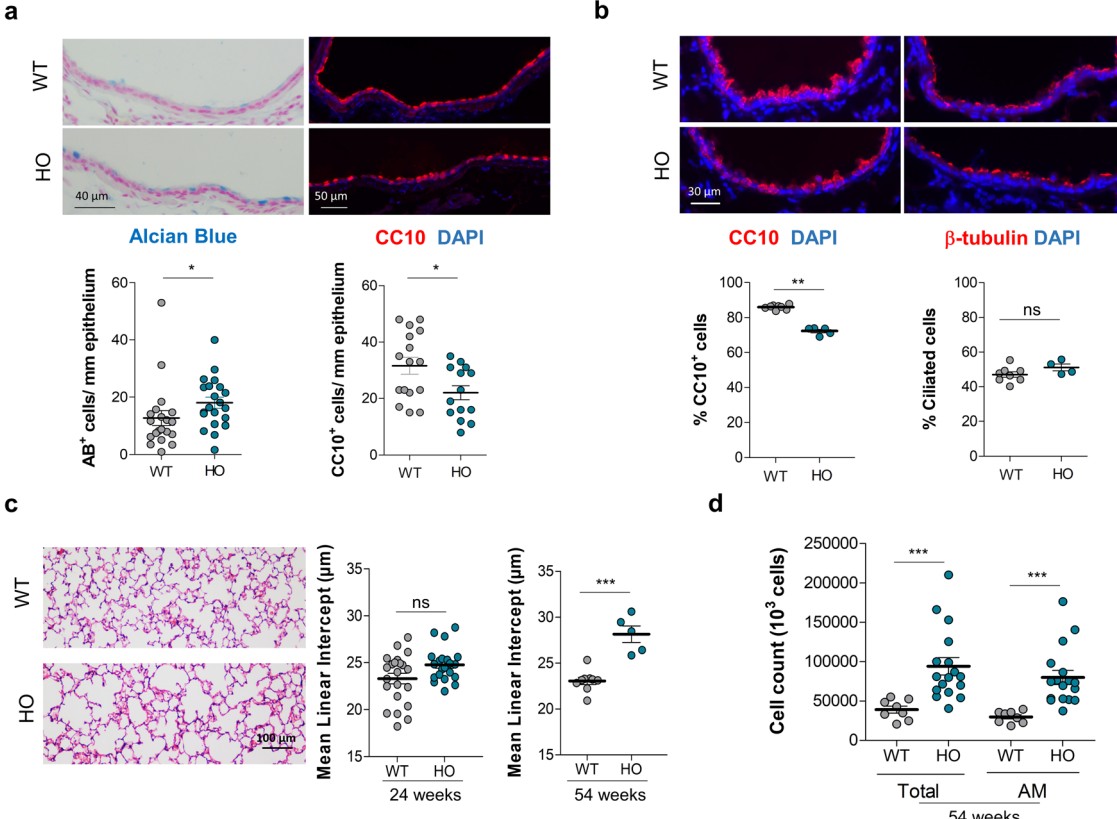

**Fig. 2 COPD-like phenotype in α5SNP mice. a** Quantification of mucus-producing cells (Alcian Blue+, n = 20 WT and n = 21 HO mice), and Club cells (CC10+, n = 16 WT and n = 14 HO mice) in proximal airways for HO (blue) and WT (gray) mice. **b** Quantification of Club cells (CC10+, n = 8 WT and n = 5 HO mice), and ciliated cells (β-tubulin+, n = 8 WT and n = 4 HO mice) in distal airways for HO (blue) and WT (gray) mice. **c** Quantification of mean linear intercept (MLI) for HO (blue) and WT (gray) 24 weeks-old (wo) mice (n = 22 WT and n = 22 HO mice) and 54 wo mice (n = 9 WT and n = 5 HO). **d** Total cell number and alveolar macrophages (AM) in BALF from HO (n = 17) compared to WT (n = 8) 54 wo mice. **a–d** Data are expressed as mean ± SEM. For statistical analyses, values are compared to the appropriate control. *p < 0.05; **p < 0.01; ***p < 0.001 (Mann–Whitney two-sided test).

the role of the α5SNP in the lung lesions, we submitted mice to an oxidative stress using cumene hydroperoxide[47–49]. We observed a significant increase of emphysema in HT and HO α5SNP compared to WT mice (Fig. 3a). In addition, there was an increase in epithelial height at the bronchiolo-alveolar junction, including bronchiolo-alveolar stem cells[50], in HT and HO α5SNP mice compared to WT (Fig. 3b). This observation was correlated with the severity of emphysema (Fig. S7). These lesions were associated with changes in pressure-volume curves attesting an alteration of respiratory function characteristic of emphysema[51] in HT and HO α5SNP (Fig. 3c, Fig. S8a). Accordingly, the alteration of lung function was confirmed in α5SNP mice by an increase in the inspiratory capacity (PVP-A), the shape constant (PVP-K) and in the static compliance (PVP-Cst), all extracted from the PV loop curve equation[52]. This was linked to an increased inflammation as revealed by a higher number of alveolar macrophages, monocytes, Tγδ lymphocytes, and invariant natural killer T (iNKT) cells (Fig. 3d, Fig. S8b). It was also associated with activation of dendritic cells, monocytes, Tγδ, and CD4+ T lymphocytes in broncho-alveolar lavage fluid (BALF) of HT and HO α5SNP mice compared to WT (Fig. 3e, Fig. S8c). Quantification of cytokines showed an increase in CXCL1, CCL5, CXCL10, and CCL20 in α5SNP mice exposed to oxidative stress (Fig. 3f).

**Epithelial repair is altered in α5SNP-expressing mouse airway epithelium.** Abnormalities in the structure, differentiation, and repair of conducting airway epithelial cells are central to the pathogenesis of chronic pulmonary disorders. Airway epithelial

remodeling is generally the consequence of repeated lesions of the epithelium with impaired repair[22]. Basal cells of the pulmonary epithelium, serving as progenitor cells, are crucial to regenerate injured epithelium and to maintain tissue homeostasis[26,27]. To assess the functional impact of the α5SNP on epithelial remodeling, we generated lentiviral (LV) vectors to express the α5SNP and WT subunits specifically in airway epithelial cells of the knock-out (KO) α5[53] mouse strain, co-expressing Green Fluorescent Protein (GFP) (Fig. 4a and b). After transduction, epithelial cells were GFP positive (Fig. 4b). As shown by CD45 and CD68 immunostaining, few mononuclear cells including macrophages were present in the submucosa (Fig. 4b), and they did not express GFP, attesting the absence of transduction by LV vectors. In vivo, epithelial repair after polidocanol instillation was altered in α5SNP-expressing mice. Specifically, the different steps of tracheal epithelium regeneration were analyzed at days 2, 6, 9, and 12 after lesion (Fig. S10a). Strikingly, regeneration in α5KO mice led to a pathological remodeling of the airway epithelium, which was more pronounced after α5SNP local transduction, when compared to α5WT transduction. Expression of α5SNP in the airway epithelium initiated a squamous differentiation program, attested by Trp63 and transglutaminase (Tgm-1) expression (Fig. S10b and c), and induced in some cases dysplastic lesions, totally absent in WT (Fig. 4c). We also observed an increased production of the pro-inflammatory cytokines TNF-α and IL-1β in α5SNP mice, in response to polidocanol, another feature described in COPD patients[54] (Fig. 4d). Importantly, pretreatment of mice with Etanercept (Enbrel®), an inhibitor of

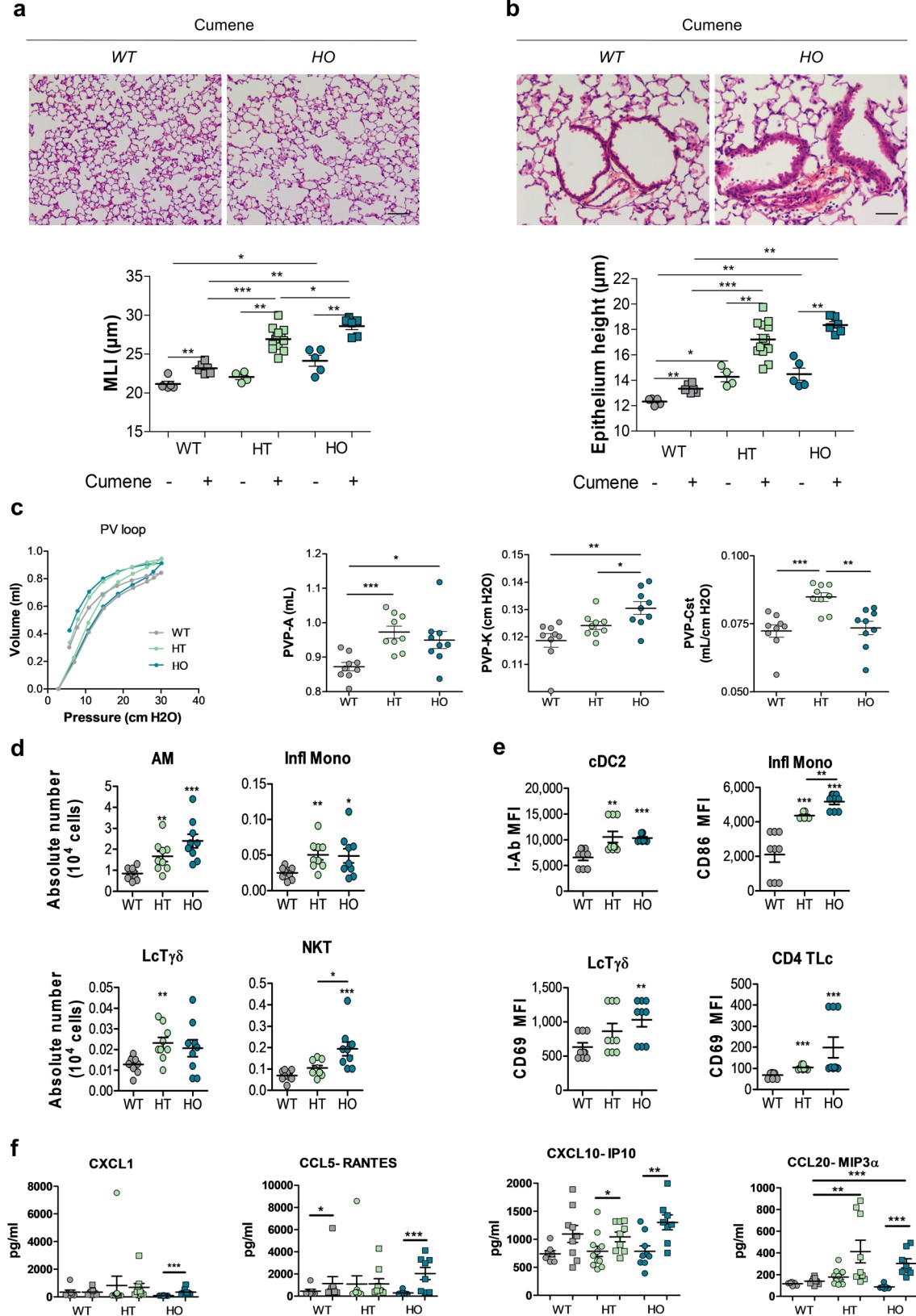

TNF-α[54], largely prevented pathological remodeling of tracheal epithelium in α5SNP mice (Fig. 4c), and NF-κB activation (Fig. 4e), confirming an important role of TNF-α in α5SNP-induced epithelial remodeling.

In vitro, using LV transduced mouse airway basal cells, we showed an increase of proliferation in α5KO cells and α5SNP

expressing cells compared to α5WT (Fig. 4f). In vitro exposure to TNF-α stimulated proliferation of these cells (Fig. 4g). Ex vivo exposure of α5KO trachea to TNF-α increased Trp63 expression, a marker of squamous cell metaplasia, IL-1β and IL-6 expression, and activated the NF-κB pathway (Fig. S11). Cell viability was controlled and we did not observe any cell death in all these

**Fig. 3 α5SNP amplified oxidative stress-induced lung inflammation and respiratory dysfunction.** Lung histology and respiratory function measurement of 14–18 wo mice, WT (gray), HT (green) or HO (blue) treated with cumene hydroperoxide (squares) or PBS (circles) (**a–c**). **a** Quantification of MLI and epithelial height at bronchiolo-alveolar junctions **b** in WT ($n = 5$), WT treated with cumene ($n = 6$), HT ($n = 4$), HT treated with cumene ($n = 13$), HO ($n = 5$), HO treated with cumene ($n = 6$) mice. Scale bars: **a** = 100 μm, **b** = 50 μm. **c** PV loop curves ($n = 16$ for each group), A (estimate of inspiratory capacity), K (shape constant) and static compliance (Cst). A and K parameters and Cst are extracted from the Salazar-Knowles equation[52] and expressed as mean ± SEM ($n = 9$ mice for each group). **d, e** Quantification of lung inflammation in BALF by cytometry ($n = 9$ mice per group). **d** Number of alveolar macrophages (AM), inflammatory monocytes (Infl Mono), Tγδ lymphocytes (LcTγδ) and natural Killer cells (NKT). **e** Activation of dendritic cells (cDC2, I-Ab MFI), inflammatory monocytes (CD86 MFI), Tγδ lymphocytes, and CD4+ T cells (CD69 MFI). **f** Expression of cytokines CXCL1, CCL5, CXCL10, and CCL20 in BALF from WT treated with PBS (circles, $n = 7$) or cumene (squares, $n = 9$), HT treated with PBS (circles, $n = 11$) or cumene (squares, $n = 9$) and HO treated with PBS (circles, $n = 9$) or cumene (squares, $n = 8$). **a–f** Data are expressed as mean ± SEM. For statistical analyses, values are compared to the appropriate non-treated control (**a, b**) or to WT mice (**c–f**). *$p < 0.05$; **$p < 0.01$; ***$p < 0.001$ (Mann–Whitney two-sided test).

experiments. To extend our findings, airway epithelial cells isolated from α5SNP patients also exhibited an increased proliferation, which likely participates in the airway epithelial remodeling (Fig. 4h). Thus, under conditions of airway epithelium stimulation, specific α5SNP expression in airway epithelial cells is associated with remodeling and secretion of pro-inflammatory cytokines.

**α5SNP decreases nAChR Ca$^{2+}$ permeability and affects basal cell proliferation and TNF-α expression through an adenylyl cyclase-dependent pathway.** To dissect the role of α5SNP in these phenotypes, we focused on the functional properties of WT and α5SNP-containing nAChRs. α5SNP substitutes aspartic acid (D) for asparagine (N) at amino acid position 398 (D398N, D397N in the mouse), located in a highly conserved site of the intracellular loop[55]. α5 containing receptors exhibit the highest Ca$^{2+}$ permeability among heteropentameric nAChRs[56], which then regulates many Ca$^{2+}$-dependent cellular processes, such as cell plasticity, growth, migration, and survival[57,58]. We established an in vitro culture system of cell-sorted basal epithelial cells (csBEC) and dissected their responses to a number of stimuli. We were able to show that α5KO or α5SNP csBEC exhibited a decrease of intracellular Ca$^{2+}$ entry upon stimulation with the endogenous agonist, acetylcholine (ACh), and also with nicotine (Fig. 5a). In parallel, we observed an increased proliferation of α5KO and α5SNP basal cells compared to WT (Fig. 5b—control conditions). Moreover, we demonstrated that this proliferation was reduced in the presence of inhibitors of adenylyl cyclase (AC), protein kinases A and C (PKA/C) or Raf (Fig. 5b), which are key elements of intracellular signaling cascades known to regulate the transcription of genes involved in proliferation[59–61], suggesting that these are some of the major pathways implicated (Fig. 5b).

We also aimed to further dissect the mechanisms responsible for the modulation of proliferation and the increased production of pro-inflammatory cytokines observed in vivo, where we highlighted a crucial role of TNF-α during wound repair. We, therefore, established a mechanical in vitro lesion model on cultured csBEC, similar to a protocol previously described by us[23]. In this model, csBEC were able to migrate to regain confluency, and TNF-α transcripts were only increased in α5SNP csBEC (Fig. 5c). Furthermore, this upregulation was partially reduced in the presence of AC and PKC inhibitors (GF109203X and SQ22536, respectively), suggesting that the same signaling pathways are involved in increased proliferation and increased expression of inflammatory cytokines in α5SNP basal cells (Fig. 5c). Importantly, only few ACs were expressed in airway basal cells, AC3 being the most represented (Fig. 5d and e). Since AC3 was the only AC exhibiting increased activity upon reduced intracellular Ca$^{2+}$ concentration[62], our results suggest that AC3 was inhibited by high intracellular Ca$^{2+}$ concentrations in WT basal cells. Importantly, AC3 was activated in α5SNP and KO

basal cells leading to increased cAMP production[63], in which we demonstrated that the Ca$^{2+}$ influx was significantly decreased. Therefore, our data suggest that activation of AC3 and enhanced cAMP drive intracellular signaling pathways implicated in TNF-α production. This is responsible for deregulated cell proliferation and an increased expression of pro-inflammatory cytokines. This would be responsible for an abnormal repair of airway epithelium and participate in pathological changes characteristic of COPD as summarized schematically in Fig. 5f.

## Discussion

In this study, we demonstrate a direct association and a nicotine- and smoke-exposure independent role of α5SNP in the development of pulmonary lesions, beyond its implication in the increase of nicotine intake[10,11]. We focused here on airway epithelial remodeling, emphysema, inflammation, impact of oxidative stress, and wound repair. In all these conditions, α5SNP was associated with pathological alterations in human samples, in mouse models, and complementary in vivo and in vitro approaches. We provide here the first functional link considering the clinical association of α5SNP with COPD.

Human airway epithelial cells and transgenic mice expressing α5SNP without any oxidative stress or nicotine intake exhibited spontaneous remodeled epithelium with an increase of goblet cells, associated with reduced anti-inflammatory Club cells and mild emphysematous lesions in mice. These observations of a mild spontaneous COPD-like phenotype were obtained in aged α5SNP mice. This propensity was confirmed in young α5SNP mice by using a model of elastase-induced emphysema. Interestingly, these lesions were also amplified by oxidative stress, provoked here by lipid peroxidation with cumene hydroperoxide mimicking the effects of pollutants. This stress also led to lung function impairment characteristic of emphysema[51], not detectable in young α5SNP mice. Inflammatory cells, such as macrophages and lymphocytes, were increased in BALF from mice exposed to oxidative stress. As indicated, after a chemical stress with polidocanol, inflammatory cells were recruited by pro-inflammatory cytokines specifically over-expressed in α5SNP mice. Consequently, these inflammatory processes likely participate in the development of emphysema and lung function impairment[64,65]. All these data show that the presence of α5SNP not only in homozygous but also in heterozygous form induces airway inflammation and alteration in pulmonary epithelium associated with an increased susceptibility to environment-related stress leading to respiratory dysfunction. It will be important to further characterize the impact of α5SNP in a model of cigarette smoke exposure, but choosing the relevant mouse strains is challenging and dissociating the carcinogenic response from the COPD-like remodeling would be difficult[66–68].

In COPD, airway epithelium is submitted to various repeated lesions due to infections, tobacco smoke, and micro-particles, which participate in airway epithelial remodeling[22]. Airway basal

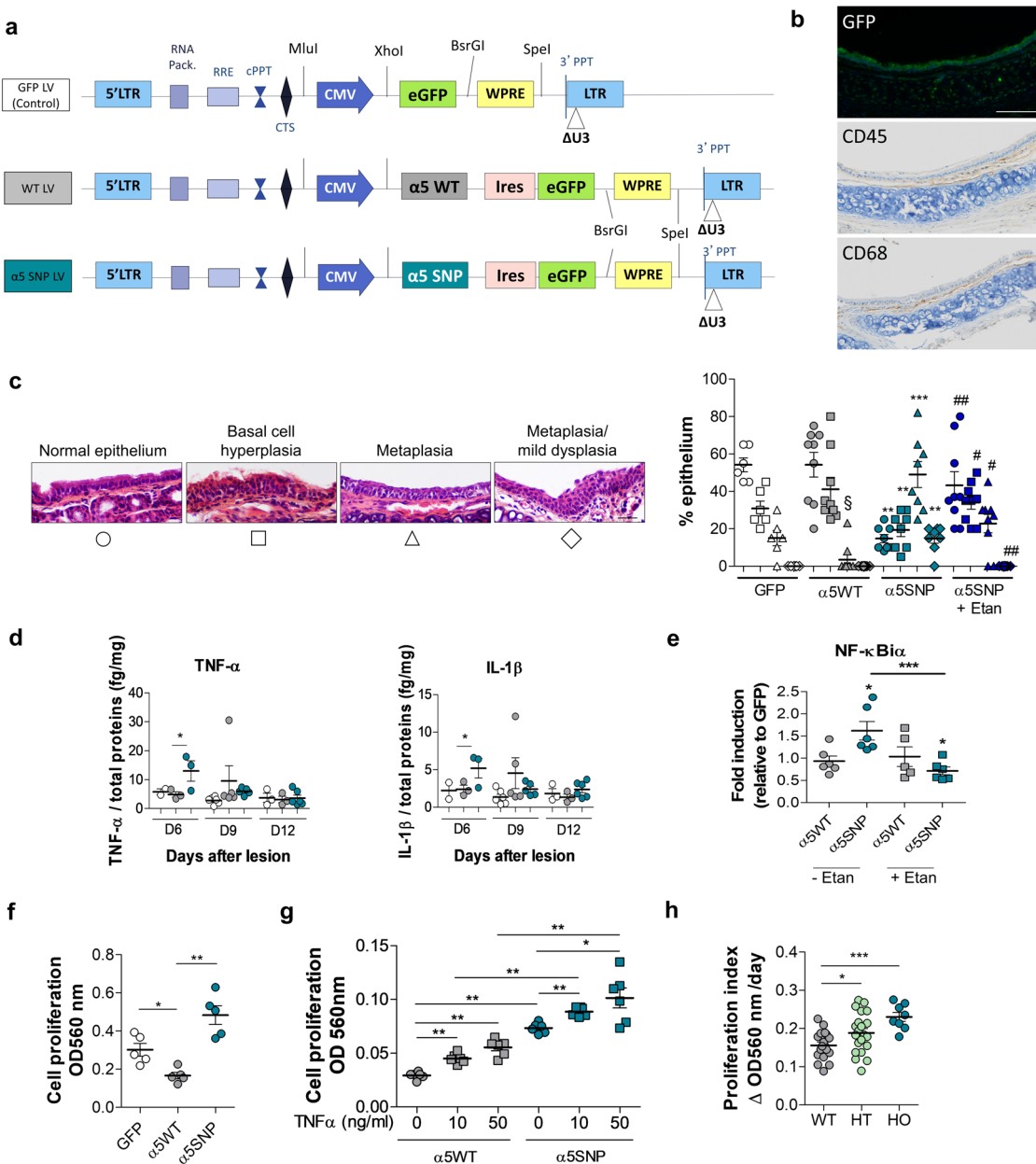

**Fig. 4 Dysregulation of epithelial repair in α5SNP-expressing airway epithelium. a** Scheme of lentiviral vectors (LV) used to locally express α5WT or α5SNP in tracheal epithelial cells in α5KO mice. **b** Top: GFP expression in tracheal epithelium. Middle and bottom: lymphocytes (CD45+) and macrophages (CD68+) in submucosa. Micrographs representative from three experiments. Scale bar: 100 μm. **c** Quantification of different epithelial phenotypes (normal epithelium, circles; basal cell hyperplasia, square; metaplasia, triangle; metaplasia/mild dysplasia, diamond-shaped) in mice with GFP LV (white symbols, $n = 6$), α5WT LV (gray symbols, $n = 9$) or α5SNP LV (blue symbols, $n = 8$) 12 days after lesion. One group of α5SNP (dark blue symbols, $n = 8$) received i.p. injections of Etanercept (α5SNP + Etan). Scale bar: 30 μm. **d** TNF-α and IL-1β production in GFP LV (white circles, $n = 6$), α5WT LV (gray circles, $n = 9$) or α5SNP LV (blue circles, $n = 8$) mice after lesion at D6, D9 and D12. **e** NF-κBiα expression in tracheal tissue extracts from α5WT or α5SNP LV mice on D6 after epithelial lesion with polidocanol (circles, $n = 6$ α5WT and $n = 6$ α5SNP LV mice) and pre-treatment with Etanercept (squares, $n = 5$ α5WT and $n = 6$ α5SNP LV mice). **f** Quantification of proliferation of tracheal epithelial cells isolated from GFP ($n = 5$), α5WT ($n = 5$) and α5SNP ($n = 5$) mice (O.D., optical density). **g** In vitro proliferation of epithelial cells isolated from α5WT or α5SNP LV mice ($n = 6$ per group) after exposure to TNF-α (squares, 10 and 50 ng/ml, 72 h). **h** Proliferative index quantification of human polyp epithelial cells isolated from WT ($n = 18$), HO ($n = 21$) or HT ($n = 8$) patients. For **c**–**h**, data are presented as mean ± SEM. Statistical analyses in (**c**) compared: WT vs GFP mice (§), α5SNP vs α5WT mice (*) and Etanercept treated vs non-treated α5SNP (#). *#,§$p < 0.05$; **, ##$p < 0.01$; ***$p < 0.001$ (Mann–Whitney two-sided test).

cells express the α5 subunit of nAChR, and as progenitor cells are responsible for the repair of airway epithelium during wound healing. Thus, they are at the origin of the epithelial remodeling. The study of wound repair in our in vivo model was characterized by an increase of squamous cell metaplasia, also observed in α5

KO mice, compared to α5WT expressing cells. This alteration is due to an over-expression of cytokines, particularly TNF-α. This was demonstrated by the prevention of remodeling by pretreatment with the anti-TNF-α drug Etanercept, and by the ability of TNF-α to induce a squamous metaplasia in mouse tracheas

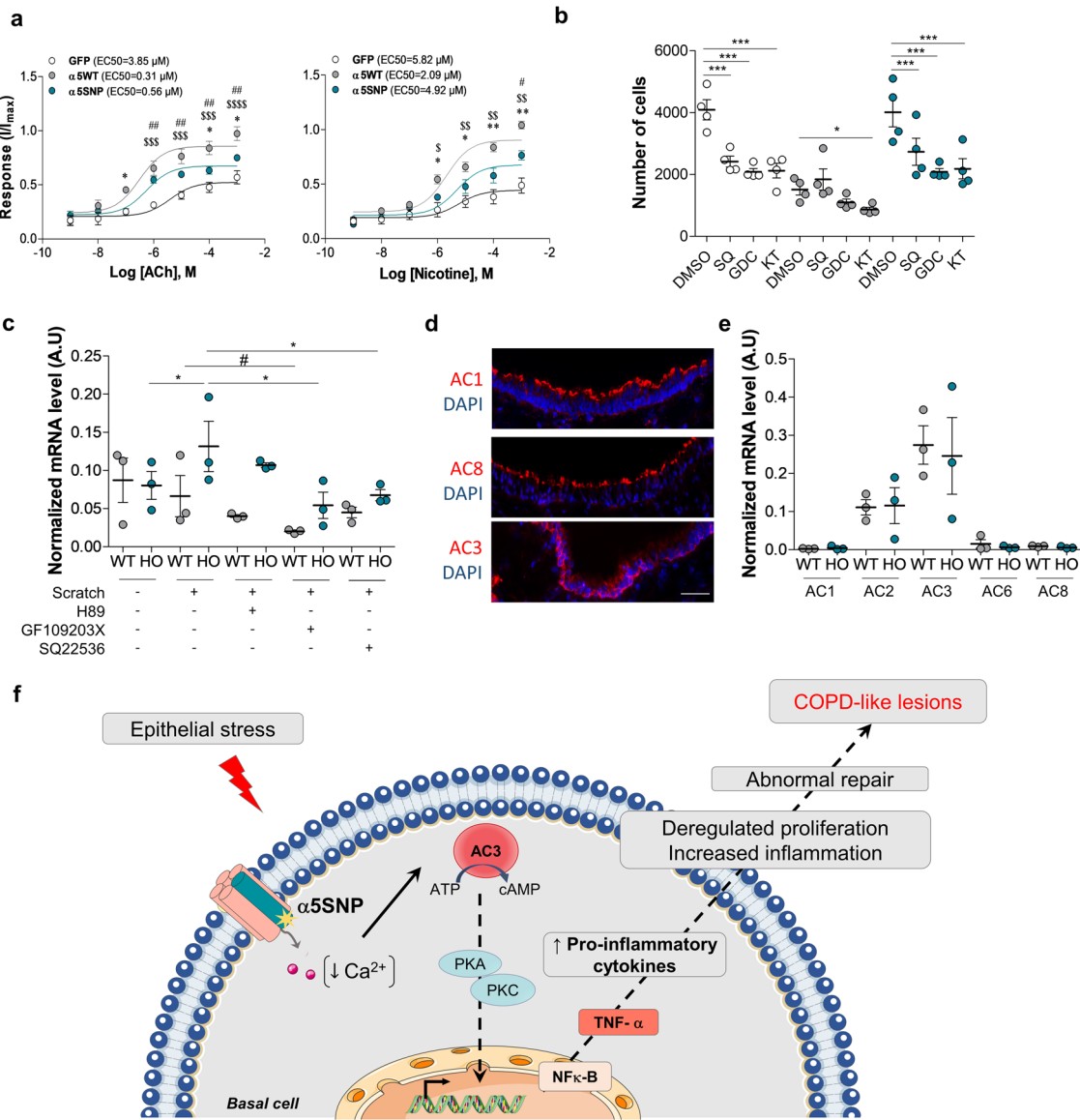

**Fig. 5 α5SNP reduces Ca²⁺ permeability and enhances basal cell proliferation and TNF-α production. a** Concentration-response curves for acetylcholine (ACh)- or nicotine-evoked changes in intracellular Ca²⁺ measured on csBEC isolated from α5WT, α5SNP or GFP LV mice. Data are expressed as mean ± SEM.The response curves for each nAChR variant are significantly different ($n = 3$ for ACh, $n = 4$ for nicotine, per variant). *$p < 0.05$, **$p < 0.01$ for α5SNP versus α5WT; $^\$p < 0.05$, $^{\$\$}p < 0.01$, $^{\$\$\$}p < 0.001$, $^{\$\$\$\$}p < 0.0001$ for GFP vs α5WT; $^\#p < 0.05$, $^{\#\#}p < 0.01$ for GFP vs α5SNP, two-way ANOVA. **b** Proliferation of csBECtreated with SQ22536 (non-selective inhibitor of ACs) or GDC0879 (Raf inhibitor) or KT5720 (inhibitor of PKA and PKC). White circles csBEC GFP, gray circles csBEC α5WT, blue circles csBEC α5SNP. Results are expressed as the number of cells determined for each group at D7 ($n = 4$ per group). *$p < 0.05$; ***$p < 0.001$; for inhibitors vs control DMSO group for each genotype, two-way ANOVA. **c** Relative expression of TNF-α normalized to cyclophilin A (arbitrary units, A.U.) after in vitro scratch assay on csBEC from α5WT (gray circles) and α5SNP mice (blue circles) ($n = 3$ per group), 15 h after scratch. *$p < 0.05$ for α5SNP scratch vs control or inhibitors (H89, selective inhibitor of PKA; GF109203X, selective inhibitor of PKC; SQ22536, non-selective inhibitor of adenylyl cyclases); $^\#p < 0.05$ for WT scratch vs control or inhibitors, Mann–Whitney two-sided test. **d** Polarized localization of Ca²⁺-sensitive AC isoforms (all in red) in tracheal epithelia from α5WT. Scale bar: 40 μm. Micrographs representative of two independent experiments. **e** Relative expression of TNF-α normalized to cyclophilin A, on csBEC from α5WT (gray circles) and α5SNP (blue circles) ($n = 3$ per group), 15 h after scratch. Data are expressed as means ± SEM of three independent experiments (**c, e**), or mean with 95% CI of four independent experiments (**b**). Data for **c, e** were quantified using the $2^{-\Delta\Delta CT}$ method[83] in log scale. **f** Scheme summarizing a mechanism of intracellular signaling in basal cells expressing α5SNP nAChR in response to epithelial stress.

ex vivo. However, under these conditions, we did not detect any goblet cell hyperplasia. Our hypothesis is that the lack of inflammation observed in tissue sections and the absence of inflammatory cells able to induce goblet cell hyperplasia[69] in this particular model may explain these observations. Of particular interest is the presence of dysplastic pre-neoplastic lesions during

the wound repair. α5SNP is also associated with lung cancer[15,16] and may also be involved in bronchial carcinogenesis in patients with COPD. All these data emphasize the essential role of α5SNP in the behavior of progenitor airway basal cells.

There are several mechanisms by which α5SNP regulates basal cell fate. We showed that α5SNP expression was associated with

decreased intracellular $Ca^{2+}$ influx in csBEC. This is in line with our previous report in HEK293T cells after heterologous expression[70]. This phenomenon can be seen as the trigger of a cascade of intracellular events involving activation of AC3, the AC with the highest expression in basal cells and sensitive to $Ca^{2+}$ levels. This leads to production of cAMP and activation of the PKA/C and Raf pathways resulting in increased cell proliferation and TNF-α production.

In conclusion, α5SNP expression, both in heterozygous and homozygous forms, directly facilitates the development of airway epithelial remodeling and emphysema, by promoting lung inflammation, jointly acting with oxidative stress. We propose to explore this functional α5SNP involvement in COPD patients considering the smoking status to complete our translational study. Understanding α5SNP contribution to airway remodeling and inflammation may help elucidating the pathogenesis and represent a target for preventing and limiting COPD[71].

## Methods

**Study design**. The objective of this work was to study the role of COPD-associated α5SNP in airway epithelial remodeling, emphysema, and inflammation. First, we investigated histological modifications in human airway epithelium on nasal polyp samples from patients undergoing polypectomy between 2016 and 2018, leading to 213 samples. Sixty-four polyps presented an epithelial abrasion higher than 70% and were eliminated from the analysis. Twenty-six polyps were collected from smokers and were eliminated from the analysis. Histological quantifications were blindly performed by two histo-pathologists independently.

Second, we characterized airway remodeling in airways of α5SNP mice compared to WT and in oxidative stress conditions. Number of mice /group/ independent experiment was approved by the Animal Ethics Committee and is available in figure legends. Mice from all our experiments were age- and sex-matched. Investigators were blinded to mice group allocation during analyses of samples. For BAL analysis, mice were excluded if blood was found in BAL samples. For mucinous cell quantification, mice were excluded if the extralobar part of the main bronchus was not disposable.

Basal cells of the pulmonary epithelium, serving as progenitor cells, are crucial for maintaining tissue homeostasis and regenerating injured epithelium[26,27]. We used a model of local expression of α5SNP specifically in airway epithelial cells including tracheal basal cells via lentiviral vectors to decipher the role of the polymorphism.

Finally, molecular events at the cellular level were investigated in vitro on FACS sorted murine airway basal cells. Experiments were performed several times and all attempts of replication were successful.

**Human nasal polyps**. This study was approved by a French Ethics Committee (Comité de Protection des Personnes Est 1—authorization 2016-A00242-49) and conducted according to the French law. Human nasal polyps were obtained from 213 patients undergoing nasal polypectomy/sinus surgery. An informed consent was obtained for each patient included in the study for the use of biological specimen and the publication of relevant clinical data including age, sex, smoking status, diagnoses. Each participant answered a survey upon their smoking status, tobacco consumption, date of smoking cessation when appropriate, and allergies/ asthma conditions. Population characteristics are available in Table S1. According to patient agreement, a polyp sample was frozen for further α5 subunit genotyping. Remaining tissue was formalin-fixed paraffin-embedded for histological analysis. Polyp HES stained slides were blindly analyzed by two histo-pathologists who scored submucosal inflammation (Fig. 1a and quantified four different epithelial phenotypes (Fig. 1b).

**Genotyping for the rs16969968 CHRNA5 polymorphism**. Polyps were ground in lysis buffer with gentleMACS dissociator (Miltenyi Biotec, MACS, Germany) and centrifuged. Supernatants were incubated overnight at 56 °C in proteinase K, and processed for DNA purification (EZ1 DNA Tissue kit) according to the manufacturer's instructions. nAChR α5 subunit coding gene CHRNA5 was amplified with DNA polymerase TaKaRa LA Taq (TAKARA Bio Inc, Japan) using primers described in Table S3. Amplification products were digested with Taq1 enzyme recognizing the following sequence: 5′-TCGA-3′, only present in the WT sequence. Digestion products were then separated by electrophoresis and gels were imaged using a LAS-1000 Imager for analysis (Aïda software, Raytest, Courbevoie, France).

**Mice**. All the experimental procedures were conducted in accordance with the guidelines of ethical use of animals from the European Community Council Directive 2010/63/EU and were approved by the Ethics Committee of the University of Reims Champagne-Ardenne (approvals 56-2012-3 and 56-2012-4), the

Institut Pasteur CETEA (01828.02), and by the French Ministry of Research (approvals 4362-2016030315538085, 2016120715475887, and 2017100512017052415271111v1). The generation of α5KO and α5SNP mice has been published[29,53]. Littermate mice were obtained by breeding heterozygous (HT) mice and genotyping (Transnetyx, Cordova, TN, USA).

**Broncho-alveolar lavage procedure and lung collection**. BAL were performed as previously described[72] with some modifications: Two BAL were sequentially performed (300 µl + 400 µl PBS for the first; 4 × 500 µl PBS for the second). Supernatant of first BAL was frozen for further analysis of secreted factors. Pellets from both BAL were resuspended in PBS for automatic total cell counting (Adam-MC cell counter, Labtech, France). Cells were "cytospun" and differential cell counts based on morphological criteria were performed on 500 cells/slide after May-Grunwald Giemsa staining (Merk-Millipore, France).

The left lobe was removed and snap-frozen in liquid nitrogen for further protein analyses after BAL procedure. The other lobes were inflated with 4% paraformaldehyde (Merk) in PBS as previously described[73] for further histological analysis. There were no inter-lobar differences in the analysis regarding to the considered biological parameters.

**Nasal cavity collection**. Nasal cavities were collected as previously described[74]. Briefly, lungs were collected after sacrifice. A solution of 4% paraformaldehyde was gently flushed in the nasal cavity which was grossly isolated. Nasal cavity was sectioned in the middle of forehead and fixed in paraformaldehyde during 24 h. Tissue specimen were decalcified for 2 days in Osteosoft solution (Merck), dehydrated and embedded in paraffin. The histological analysis was performed on nasal airway sections after HES staining. Alteration of epithelial phenotype and inflammation was estimated as described for lung sections.

**Ex vivo cultures of murine tracheas**. Tracheas were collected, longitudinally opened, and exposed during 48 h to TNF-α, IL-1β, IL-6, or PBS (control) in RPMI-HEPES medium with 1% penicillin–streptomycin. After stimulation, tracheas were snap frozen and conserved at −80 °C for RNA and protein analysis.

**Immunohistochemistry**. Immunofluorescence was performed on FFPE sections of murine lungs with primary antibodies listed in Table S2. Adenylyl cyclase and cytokeratin immunostainings were performed on 8 µm frozen sections of mouse trachea embedded in OCT Compound (Tissue-Tek, Sakura, Villeneuve d'Ascq, France). Slides were then incubated with DAPI (2 µg/ml, D9542 Sigma-Aldrich, France) and a secondary antibody (Table S2). Microscopy was performed with an Imager Z1 epifluorescence microscope (Zeiss, Oberkochen, Germany) equipped with CoolSNAP HQ camera and ZEN software (Zeiss). For CD45 and CD68 detection, immunostainings were performed using Ventana Benchmark XT autostainer (Ventana Medical Systems, Inc., Tucson, AZ, USA). Subsequent steps were performed with the ultraView universal DAB detection kit (Ventana). Microscopy was carried out using a Nikon DS Fi2 camera and NIS-Elements software (Nikon, France).

**Emphysema quantification**. Five µm sections of FFPE lungs were HES-stained. For each slide, three pictures of parenchyma per lobe and per mouse were taken using a Nikon DS Fi2 camera and NIS-Elements software (Nikon). An automatic quantification of emphysema was performed by image analysis using ImageJ software (NIH, USA) with a macro developed in our laboratory. Briefly, acquired images in RGB (Red, Green, Blue) color space were represented in HSB (Hue, Saturation, Brightness) color space. The saturation parameter was used to discriminate lung tissue from air after binarization. This classification is refined and quantified using the particle analyze function in ImageJ software, one particle corresponding to an airspace. Airspace sizes inferior to 5 µm diameter were ignored. For each image, this macro gave the mean airspace surface, airspace perimeter, and the count of airspace per image. The Mean Linear Intercept (MLI) was also calculated on these images using another macro developed in our laboratory. The previously binarized image was vertically and horizontally scanned (step: 1pixel) to determine the height and width of each airspace. MLI was calculated as the mean of all measures per image.

**Alcian Blue staining and mucinous cell quantification**. FFPE lung sections were deparaffinized and incubated during 1 h in 1% Alcian Blue 8GX solution in 3% acetic acid (Sigma). A counterstaining was performed using Nuclear Fast Red solution (Sigma) during 15 min. Microscopy was performed using a Nikon DS Fi2 camera and NIS-Elements software (Nikon). Positive cells for Alcian Blue staining ($AB^+$) were manually counted in the main bronchi (Fig. S2) and results were expressed as cells/mm epithelium.

**Measurement of lung function**. Fourteen to eighteen weeks-old male WT mice or HT or HO for α5SNP were either nasally instilled with 50 µl of either PBS or 3 mg/ ml cumene hydroperoxide. Nasal instillations were performed 7, 5, and 3 days before lung analysis. Anesthetized mice were tracheotomized and connected via an 18G cannula to a flexiVent FX system operated by the flexiWare software v7.7

(SCIREQ Inc., Montreal, QC, Canada). Animals were ventilated at a respiratory rate of 150 breaths/min and mechanical properties of the subjects' respiratory system were assessed at baseline, i.e. before the construction of a full-range PV curve[75]. The pressure-volume curve assesses the distensibility of the respiratory system at rest over the entire inspiratory capacity. The deflation arm of this curve is fitted with the exponential function described by Salazar and Knowles[52]. Static compliance (Cst), the parameters A (estimate of inspiratory capacity) and K (shape constant) can be extracted from the Salazar–Knowles equation. Static compliance (Cst) reflects the intrinsic elastic properties of the respiratory system (i.e., lung + chest wall).

### Flow cytometry

*BAL cell analysis.* Cells were counted by Turks staining and incubated with an appropriate panel of antibodies (Table S2) for 30 min in PBS 2% FCS. Data were acquired on a LSR Fortessa (BD Biosciences) and analyzed with FlowJo™ software v7.6.5 (Stanford, CA, USA). Gating strategies are illustrated in Fig. S12. Alveolar macrophages are characterized as (CD11c+, F4-80+, Siglec F+), cDC2 dendritic cells (CD11c+, F4-80−, I-Ab+, CD11b+), inflammatory monocytes (CD11c−, F4-80+, Ly-6C+, CCR2+, CD64+), Tγδ lymphocytes (CD5+, TCRγδ+) and CD4+ T cells (TCRαβ+, NK1.1−, CD4+).

*Mouse tracheal basal cell FACS sorting procedure.* eGFP+/NGFR+ (transduced α5KO mice) or NGFR+ (WT and α5SNP KI mice) basal cells were sorted on FACS ARIA III with FACS Diva software (BD Bioscience)[27] (Fig. S13). Airway basal cells were collected in MTEC proliferation medium[76], and cultured immediately for calcium imaging, cell proliferation, and in vitro wound scratch assays.

### Measurement of epithelium height

Pictures were taken for all the bronchiolo-alveolar regions identified on HES-stained mouse lung FFPE sections. Quantification of the epithelium height was computed using ImageJ software and a plugin developed in our laboratory. The epithelium basal side was manually delimited using the ImageJ segmented line selection tool and a region of interest was determined by an automatic detection for the apical side. The surface of the resulting ROI was calculated and the mean height of the epithelium was determined from the length of the basal pole.

### Lentiviral vector construction and mouse transduction

The lentiviral (LV) expression vectors were derived from the pHR' expression vectors first described by Naldini et al.[77] with subsequent modifications. In this work we used the second generation of lentiviral vectors[78].

*LV pCMV-α5-IRES-eGFP and pCMV-α5SNP-IRES-eGFP construction.* PGK-α5-IRES-GFP and PGK-α5SNP-IRES-GFP vectors coding respectively for the WT or the mutated nicotinic α5 subunit[10] were modified by replacing the mouse phosphoglycerate kinase (PGK) promoter by the human cytomegalovirus (hCMV) promoter to obtain a stronger expression in airway epithelial cells. Briefly, we used the pre-existing pTRIP-hCMV-eGFP LV (a gift of the LV network of the School of Neuroscience of Paris, ENP) to generate a DNA fragment containing the CMV sequence by using BssHII and XhoI restriction sites. This fragment was then ligated into the pre-existing PGK vectors between the same restriction sites. Production and titration of LVs was performed using standard procedures[29,79].

### LV transduction

Eight to ten weeks-old α5KO mice were transduced with LV by tracheal instillation[80]. A first instillation of 15 µl of 1% L-α-lysophosphatidylcholine in PBS (LPC, L4129, Sigma) was carried out to pre-condition the airway epithelium and rendering it permissive for the infection of basal cells by LV[81]. Thirty minutes later, 20 µl of a 10[7] TU/ml LV solution were instilled into the trachea.

### Polidocanol-induced lung lesions

Epithelial stress and desquamation were induced in mouse trachea after instillation of 2% polidocanol in PBS (Sigma-Aldrich, 32 µl) as previously described[42]. At D3, D6, D9, and D12 after tracheal damage, global aspiration was performed (700 µl PBS) and the supernatant was frozen for further analysis of secreted factors. Tracheas were collected and conserved at -80 °C for further analysis or were formalin-fixed paraffin-embedded for histological analysis. Etanercept (Enbrel®, Pfizer) was injected i.p. in α5SNP mice (10 mg/kg) four days and 2 days before polidocanol instillation, the day of the instillation and 2 days later. The histology of tracheal epithelium was blindly analyzed on HES stained slides by two histo-pathologists who identified and classified different epithelial phenotypes (Fig. 4c, left). Each epithelium phenotype was expressed as percentage of the analyzed tracheal epithelium.

### Emphysema induced by purified pancreatic elastase

We included 18 mice (15–21 weeks) treated with intra-tracheal instillation of PBS or 0.1 unit of PPE (Sigma-Aldrich E-1250), in 60 µl of PBS in 10 WT mice (3 PBS vs 7 PPE) and 8 α5SNP KI mice (3 PBS vs 5 PPE). Mice were sacrificed 21 days after instillation and emphysema was quantified as described above.

### Cytokine quantification

Frozen tracheas were ground using an Ultra-Turrax (Janke & Kunkel, IKA, Germany) in Tissue Extraction Reagent I (Invitrogen, Carlsbad, CA, USA.) with protease inhibitor cocktail Complete Mini (Roche Diagnostics, France). TNF-α and IL-1β assays were carried out using the CBA Mouse Enhanced Sensitivity Kit (BD Biosciences) according to the manufacturer's instructions. Results were acquired using a BD LSR Fortessa™ cytometer (BD Biosciences) and BD FACSDiva™ software (BD Biosciences) and processed using FCAP Array Software 3 (BD Biosciences). Results were normalized to total protein concentration determined by microplate DC protein assay (Bio-Rad, Hercules, USA) according to the manufacturer's instructions.

### Cell culture and proliferation tests

*Human airway epithelial cells (HAEC).* Cells were isolated from human nasal polyps and cell proliferation was evaluated by MTT assay (1 mg/ml, M-2128, Sigma) as previously described[23,42]. $OD_{560\,nm}$ was read in a Multiskan Ex spectrophotometer (Thermo Scientific) and plotted for every day to calculate a proliferation index from the slope of the curve, allowing standardized quantitative comparisons.

*Murine epithelial cells.* Cells were isolated from trachea similarly to polyps with some modifications. Tracheas were digested in 0.05% pronase and cells were seeded in presence of 20% FCS during the first 48 h. After 10 days, cell proliferation was evaluated by MTT assay.

*Cell-sorted basal epithelial cells (csBEC).* Tracheal basal epithelial cells from transduced α5KO mice were isolated and cultured as described in ref. [82]. After 4 days cells were sorted on a FACSAria™ III cell sorter (BD Biosciences), see Fig. S13, and seeded in presence of 10 µM SQ22536 (non-selective inhibitor of adenylyl cyclases) or 10 µM GDC0879 (selective inhibitor of Raf) or 10 µM KT5720 (inhibitor of PKA and PKC) in MTEC proliferation medium. All inhibitors were solubilized in DMSO and the same % of DMSO was added to the control condition. After 7 days, cell proliferation was assessed using the Opera QEHS system (Perkin Elmer Technologies, Waltham, USA) to quantify the total number of GFP + cells per well. Two replicates per condition were imaged.

### Measurement of intracellular $Ca^{2+}$

csBEC were cultured on collagen type I (5 µg/ml, Sigma). Changes in the cytosolic concentration of free $Ca^{2+}$ were measured using the $Ca^{2+}$ indicator Rhod 4-AM (AAT Bioquest, 21120) when cells reached 60% confluence. Cells were pretreated with 10 µM α-bungarotoxin (αBgt, Tocris, 2133) for nicotine-evoked measures or with 10 µM αBgt, 10 µM hemicholinium-3 (HC3, Sigma-Aldrich, H108) and 20 µM atropine (Sigma-Aldrich, A0132) for acetylcholine-evoked measures (Fig. S14). After pre-treatments, cells were washed twice with MTEC proliferation medium, incubated with 3 µM Rhod 4-AM preparation containing 0.01% of Pluronic F-127 (ThermoFisher, P3000MP), for 45 min at room temperature in the dark, then washed twice with PBS without $Ca^{2+}$ (ThermoFisher, 14190094), and placed at 37 °C in a 5% $CO_2$ incubator for 10 min before initiating the assay. Ten cells per condition were imaged on a spinning-disk confocal microscope (Revolution XD, Andor), with a 40x oil 1.3 NA objective. A single Z-plane was imaged. Cells were maintained at 37 °C during imaging.

Following a 1 min baseline record, agonist was manually added onto each sample and fluorescence was recorded at 200 ms intervals for 5 min. At the completion of the agonist stimulation, 1 µM ionomycin was added to trigger a maximal $Ca^{2+}$ flux and fluorescence was recorded for 10 s at 200 ms intervals. For each movie, a region of interest (ROI) was drawn over each cell in the field by using ImageJ software (NIH, Bethesda, MD, USA), and the mean intensity was plotted over time for each ROI. Agonist responses were normalized by dividing the maximal peak value for the agonist-stimulated fluorescence (F) (maximal peak agonist-stimulated fluorescence–baseline fluorescence) by the maximal peak fluorescence value (Fmax) triggered in the presence of 2 µM ionomycin.

### In vitro scratch wound repair assay

csBEC were cultured in MTEC medium until confluence. Then, the monolayer was scraped in a straight line to create a scratch using a 10 µl pipette tip. Cultures were then treated with 2 µM H89 dihydrochloride (selective inhibitor of PKA) or 10 µM GF109203X (selective inhibitor of PKC) or 10 µM SQ22536 (non-selective inhibitor of adenylyl cyclases). After 15 h, cells were harvested for mRNA extraction. Two replicates per condition were analyzed.

### qRT-PCR on csBEC

Total RNA was extracted from csBEC using a RNeasy mini kit (Qiagen, 74104). cDNA was synthesized with M-MLV reverse transcriptase (Invitrogen, 28025-013) and random hexamer primers (ThermoFisher, SO142). qPCR was carried out in a LightCycler 480 instrument (Roche Applied Science) using the KAPA SYBR FAST qPCR Master Mix (KapaBiosystems, KK4611) and specific primers detailed in Table S3. All reactions were run in duplicate and results were normalized to cyclophilin A expression.

**qRT-PCR on ex vivo tracheas**. Total RNA from transfected mouse trachea was isolated using the High Pure RNA isolation kit (Roche Diagnostics) and lyophilized. 500 ng of total RNA was reverse-transcribed using the Transcriptor First Stand cDNA Synthesis kit (Roche Diagnostics). qPCR was performed using the fast Start Universal Probe Master kit and LightCycler 480 SYBR Green I Master according to the manufacturer's instructions in a LightCycler 480 Instrument (Roche Diagnostics). The specific primers are described in Table S3. Results were normalized to Hprt expression.

**Statistical analysis**. Statistical analyses were computed using GraphPad Prism software (versions 5 and 7, GraphPad software, San Diego, USA) in accordance with the size of samples and the nature of the experiment. For in vivo and in vitro studies, values were compared to the controls with a bilateral and non-parametric Mann–Whitney test, or with a one-sample $t$-test, or a two-way ANOVA test when appropriate. Statistical significance was accepted for an error risk inferior to 5% and is represented as follows: $*p < 0.05$; $**p < 0.01$; $***p < 0.001$.

Concerning the polyps: after exclusion of samples presenting more than 70% of epithelial abrasion and originating from smokers, 123 patients were selected. Epithelial outcomes were analyzed as binary phenomenons: inflammation (low for scores 0 and 1; high for scores 2 and 3); secretory cell hyperplasia (low if <10%; high if ≥10%); proportion of normal epithelium (low if <50%; high if ≥50%). First, a univariate analysis evaluated the effect of covariates (sex, genotype, and allergy/asthma) on epithelial phenotypes and inflammation by a $\chi^2$-test and the effect of age by a Student $t$-test (Table S1). Second, covariates with $p$-value < 0.1 in univariate analysis were entered into a logistic regression model for multivariate analysis. Odds ratios and corresponding 95% confidence intervals (CI) were determined. A $p$-value < 0.05 was considered statistically significant in multivariate analysis. The statistical analyses were computed using Stata software (Version 10.0, Stata Corporation College Station, TX, USA).

**Reporting summary**. Further information on research design is available in the Nature Research Reporting Summary linked to this article.

## Data availability
The authors declare that all data supporting the findings of this study are available within the article and its Supplementary Information files. Any data will be available from the corresponding authors upon reasonable request. Source data are provided with this paper.

## Code availability
Emphysema and epithelium height were quantified using custom codes in ImageJ v1.51J software. They may be available to researchers interested in access to the data upon reasonable request with the corresponding authors.

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

## Acknowledgements

The authors would like to thank Kamel Maouche for his contributions, Philippe Noël, Marin Moutel, Ioana Ciupea, Leo Olory-Garnotel, Martine Soudant, Laurène Schlick, Arnaud Bonnomet (Plateforme d'Imagerie Cellulaire et Tissulaire PICT-URCA), and Sandra Audonnet (Plateforme de Cytométrie en flux URCACyt) for their technical help, Gaëtan Deslée and Giorgia Egidy for comments on the manuscript, the UtechS Photonic BioImaging ("Imagopole"), part of the France-BioImaging infrastructure supported by the French National Research Agency (ANR-10-INSB-04-01, Investments for the Future), for the use of the Opera system, and the Technology Core of the Center for Translational Science (CRT) at Institut Pasteur, in particular Pierre-Henri Commere from the flow cytometry platform for cell sorting experiments. This work was supported by the French *Agence Nationale de la Recherche* (ANR, program Nicopneumotine to U.M.), the French *Institut National du Cancer* (INCa, program *Tabac 2017* to U.M., and BIO-SILC), the *Ligue contre le Cancer Région Grand-Est* (to J-M.T.), the *Association de Recherche sur le Cancer* (ARC, program *Equipes ARC* to P.B.), the CHU of Reims (research program RINNOPARI to P.B.), the *Ville de Reims* (to J.R.), the *Prix de la Chancellerie Sorbonne, Legs Poix* (to U.M.), Fondation pour la Recherche Médicale (program EQUIPE 2019, to U.M.) the Lions Club of Soissons and Crépy-en-Valois (to P.B.), the Institut de Recherche en Santé Publique (IReSP, program PINACRAECOPD, to P.G.), Institut Pasteur, Inserm, CNRS, and the National Institutes of Health (grants CA089392 and DA015663 to J.A.S.). U.M. is a member of the Laboratory of Excellence, LabEx BIO-PSY. As such, this work was supported by French state funds managed by the ANR within the *Investissements d'Avenir* program under reference ANR-11-IDEX-0004-02.

## Author contributions

U.M. conceived the study, and with V.Do., J-M.T. and P.B. supervised the work. U.M., P.B., J-M.T., and M.L. obtained funding. P.B., U.M., V.Do., and J.R. wrote the paper. J.R., J-M.T., S.P., M.L.F., N.L., P.G., M.P., G.K., and V.Do. carried out experiments and analyzed data. V.Do. and P.B. carried out histopathological analysis. J.A.S. generated and contributed an unpublished mouse line. J-C.M. provided nasal polyps and clinical information. V.Da. performed statistical analysis. J.C. and A.J. developed ImageJ macros and plugins.

## Competing interests

The authors declare no competing interest.
