## [Peer Review File · Nature Communications]

REVIEWER COMMENTS

Reviewer #1 (Remarks to the Author):

In this study by J Routhier, et al., the authors investigate the consequences of rs16969968 (a5SNP), a SNP that has been associated in several GWAS with COPD, lung cancer, and cigarette smoking behavior. The authors report a comprehensive study examining the smoking independent pathologic changes in human nasal epithelial tissue and demonstrating the multimodal functional consequences of a5SNP in a transgenic mouse model with multiple environmental triggers. The authors also put forward a rational mechanism and pathways involved in the observed pathophysiology using cultured airway epithelial cells. Overall, the authors provide a robust study elucidating a causal pathway in the development of COPD, including a possible drug target and therapeutic option. While this study has several prominent strengths, we have several issues that we feel should be addressed prior to publication.

1) Human association data:

a) The identification of the locus and causal variant (page 3) is confusing, e.g. "chromosome 15q25, a human haplotype encompassing three genes". The haplotype is not the same as the locus, and there are many more genes than the three mentioned. In fact, there is evidence about IREB2 as an effector gene at this locus, and also for more than one signal. There is no discussion of the extensive literature on fine mapping, identifying the causal variant, etc -- while much of this data supports investigation of the nonsynonymous SNP, the lack of mention is potentially misleading about the complexity of this locus.

b) The idea that "solely 20-50% are tobacco smokers" is a concerning and inaccurate statement. Certainly one could identify a population where the smoking prevalence is very low, but studies that have identified this locus for COPD have not studied these populations, and from a public health standpoint this statement could very easily be misconstrued.

c) The human genetic evidence requires additional discussion. The largest and most recent study including the UK Biobank by Shrine et al does not identify any effect of this locus on lung function in non-smokers. This finding seems to argue that there may be issues in the relevance of their cell type selection and the relevance of the mouse model to human disease, and also raises the question whether issues in accurately ascertaining smoking history could have led to the identification of an association in 'non-smokers' or after adjusting for smoke exposure.

d) The analysis of the human subject is problematic (page 5, paragraph 1, and others). The genetic data does not account for ancestry and possible population substructure - the allele frequency varies greatly by population. In addition, the association with cigarette smoking exposure makes controlling / adjusting for this covariate absolutely critical. Dividing up smokers into three somewhat arbitrary groups likely underestimates the contribution of smoking -- in fact, using their three groups of smokers, there appears to be no effect of smoking on epithelial inflammation, and smokers appear to have similar amounts of normal epithelium, which brings into question whether their smoking ascertainment or analysis is correct (Table 1 and Supplementary Table 1). Use of current vs former, duration, and packs per day (or pack-years) are standard approaches for adjustment and should be used.

The authors also performed a univariate screen to select covariates to include in a multivariable logistic regression to determine the effect of a5SNP genotype on odds of inflammation. While univariate screen should pick up most relevant confounders, there may be interaction effects that may not appear without other variables. The authors have a large enough sample size to consider including the relevant available covariates without concern for overfitting, and there do not appear to be any obviously colinear variables.

2) Cell types:

The authors use nasal polyps as a proxy for airway epithelium. basal cells..." It is understandable and

justifiable why they chose nasal polyps, but there are substantial differences between different types of airway epithelium, and identification of the causal or effector cell types for the pathology remains unanswered. Similarly, (page 14, paragraph 3) they used tracheal basal cells -- "we used a model of local expression...including tracheal. Perhaps this is an experimental limitation of murine models, but it is unclear why the authors focused on tracheal pathology in assessing the role of $\alpha 5$ SNP. While there are hallmark tracheal changes sometimes seen in COPD (ie submucosal hyperplasia), the airway changes in COPD are typically seen more distally (<https://doi.org/10.1186/s12931-019-1017-y>). The authors should provide further rationale for the use of tracheal samples for analysis rather than more distal airway tissue.

3) Mouse data:

- a) In some situations, the heterozygous data is presented, and in other cases, only the homozygous variant is presented. Is there a justification for excluding the hets if they are available? Similarly, their prior study used male mice, but in this study says "mice were... sex-matched"? More information on the sex balance would be helpful.
- b) Given the substantial effects of this gene on behavioral phenotypes described in their previous work, is it possible that some of the effects they see are modified by behavior -- for example differences in socialization and effect on stress, diet and effects on nutrition / weight? Were these behavioral effects measured and accounted for?
- c) Page 36, Figure 2: they would consider the analysis of the age effect different between the two groups via test for an interaction term (see: <https://www.nature.com/articles/nn.2886>).

4) COPD phenotypes:

- a) Page 3, paragraph 3: "Other than emphysema..." May be better stated as a subtype of COPD or in addition to emphysema.
- b) Page 4, paragraph 1: "...COPD lesions..."
- c) Page 6, paragraph 2: "...an increase in the inspiratory capacity..." this is not a hallmark of lung function changes in COPD; humans with COPD have decreased inspiratory capacity owing to increased functional residual capacity via increased residual volume
- d) Page 10, paragraph 1: "...COPD-like lesions..." Probably best to define this using more commonly used terms – emphysema vs chronic bronchitis – or pathologic changes characteristic to COPD
- e) Page 16, paragraph 3: "The cardiac pulmonary lobe..." Classically the upper lobes/zones are the most involved in COPD, which makes the choice of a "middle" lobe questionable. The authors nonetheless demonstrate pathologic changes, which may make the choice moot.

Additional comments:

- a) 3) Page 8, paragraph 1: "Importantly, pre-treatment of mice with Etanercept..." There is no mention of this experiment in the methods section. While demonstration of translation of findings to therapeutics is important, the authors should include discussion of rationale and transparency regarding choice of drug. In general, given the extensive number of experiments, a diagram or table of the experiments could be helpful.
 - b) Page 8, paragraph 1: How do the authors account for no increase in TNF- α in BALF of $\alpha 5$ SNP mice but increase in TNF- α in $\alpha 5$ KO mice following $\alpha 5$ SNP transduction?
 - c) Page 7, paragraph 1: "All these findings..." This sentence is a conclusion and does not belong in the section reporting results
 - d) Page 8, paragraph 2: "Thus, under conditions of airway epithelium aggression..." Aggression does not seem to be the correct word to use in this case. Perhaps stimulation or agitation would fit better.
- Page 11, Paragraph 1 -- what is meant by " $\alpha 5$ SNP intervened"?

- e) Page 12, paragraph 2: "...a frequent transient prolonged squamous cell metaplasia." This phrase is unclear and transient prolonged is a bit of an oxymoron.
- f) Throughout: the use of HT and alpha5SNP as terms for the heterozygote and the homozygous models is confusing (see, for example, Supplement, Page 2).
- g) I find it hard to interpret the video - there is no control group, and no description of what we are seeing.
- h) Page 24, Paragraph 2 -- what were the reasons for dichotomization of the quantitative or ordinal scales?
- i) Page 35, Figure 1: 'Chorionic' is not a commonly used term outside of fetal anatomy, and is somewhat confusing -- is there an alternative that can be used?
- j) Page 36, Figure 2: Need to clarify the time period for experiments (are they 24 weeks, except as noted)?
- k) In general, it would be much more informative to plot the individual data points, rather than box plots (particularly those with wide confidence intervals), e.g. <https://journals.plos.org/plosbiology/article?id=10.1371/journal.pbio.1002128>
- l) The PV curves for the heterozygotes appear to have a higher compliance than the wild type and the alpha5SNP -- can the authors explain this phenomena?

Reviewer #2 (Remarks to the Author):

This is an interesting study on the role of the alpha5 nAChR in COPD. Although well conducted, the study needs some more work.

It has been shown that similar to chromaffin cells in the adrenal gland airway epithelial cells express multiple subtypes of nAChRs (including the alpha5) that regulate the secretion of catecholamines which function as autocrine growth factors for airway epithelial cells and lung adenocarcinomas derived from them, a response mediated by binding of the catecholamines to adrenergic receptors that stimulate the activation of AC resulting in the formation of intracellular cAMP. All nAChRs are cation channels which depolarize the cell membrane in response to the influx of cations. Even though in the present study the preferential influx of calcium through the channel of the alpha5nAChR was reduced by the SNP, membrane depolarization obviously still occurred likely via compensatory increase in the influx of non calcium cations as evidenced by the observed increase of AC. nAChRs do not directly increase AC but only obtain this response in an indirect fashion via the nAChR-induced release of catecholamines.

The authors need to additionally measure catecholamine release in their experimental systems in order to obtain a correct interpretation of their data.

Reviewer #3 (Remarks to the Author):

Polymorphisms in the CHRNA3 and CHRNA5 genes have been found and replicated many times in genome wide association studies for lung function in the general population. The general assumption has been that these polymorphisms relate to smoking addiction and some data in the literature support this contention, whereas other data seems to support this to be a smoking-independent risk factor. In fact, several studies have argued, that nicotinic receptors have additional roles in peripheral tissues (beyond neurotransmission) in governing inflammation, epithelial cell proliferation and differentiation and others. For the most part, these studies have addressed the role of the alpha7 nAChR in such responses. The role of the alpha5 nicotinic receptor has been studied mostly in the context of lung cancer development. The current work shows a major role for CHRNA5 and for

polymorphisms of CHRNA5 specifically, in the regulation of epithelial cell biology in the nose and lung. This is a major finding, and in fact long-awaited data, that will have a major impact on the field. Nonetheless, I do have some concerns with the data presented and the data interpretation that I would like to discuss.

1. I had difficulties connecting the nasal polyposis data in humans to the goblet cell hyperplasia data in the intrapulmonary airways of the alpha5SNP mice. In this respect, I recommend the following changes:
 - a. Please do not refer to nasal polyposis as a COPD-like condition (Figure 1 title) as it is not. It does provide rationale for a role of CNRNA5 in epithelial abnormalities and I suggest to define it as such.
 - b. The paper would benefit enormously from data linking CHRNA5 genotype to epithelial cell characteristics of the intrapulmonary airways. This would make the work much more relevant to COPD.
 - c. In the absence of such data, the context of the human data should be rephrased and more carefully positioned. Furthermore, nasal epithelial data in the mouse would be very helpful in that case to better link the human and the mouse data.
 - d. The polidocanol data is interesting; however I disagree with the assertion that this is a COPD model. It is an upper airway injury model and valuable in its own right, but does not model COPD.
 - e. If human COPD data cannot be obtained, however, at least in the opinion of this reviewer, the title needs to be adapted to reflect that the work is on the role of alpha5 nicotinic receptor on nasal and airway epithelial remodeling and not on COPD. There is no COPD data in the paper.
2. Presumably, HT means heterozygous but I could not confirm this in the paper.
3. The authors have the alpha5KO mice available but I could not find any data in the paper on the phenotype of these mice in comparison to WT mice. Is endogenous expression of alpha5 sufficient for the epithelial cell changes observed, or is overexpression of the channel protein (either local in epithelial cells or whole body) required to expose the effects? In my opinion, the KO mouse could be valuable in assessing this important question, for example by using a cigarette smoke exposure model which would directly link nicotine exposure to pulmonary changes as a function of the absence or presence of the alpha5 receptor.
4. It is unclear to me what WT means in Figure 2/3. Are these wild-type mice or mice overexpressing the WT alpha5 channel protein? If they are wild-type mice, how can the authors deduce receptor overexpression effects from SNP-specific expression effects?
5. There is one major issue I cannot get my head around. The minor A/A allele produces a less efficacious channel than the WT allele. Yet, in the present study, only expression of the minor allele produced functional effects in the alpha5 KO mice. Mice expressing the more effective alpha5 WT allele were unaffected in comparison to alpha5 KO mice expressing GFP. This does not make sense to me: how can absence of expression and overexpression of the WT receptor have similar effects with overexpression of a less active receptor having differential effects? This discrepancy needs to be adequately explained and supported by data.

REVIEWER COMMENTS

Reviewer #1 (Remarks to the Author):

In this study by J Routhier, et al., the authors investigate the consequences of rs16969968 (a5SNP), a SNP that has been associated in several GWAS with COPD, lung cancer, and cigarette smoking behavior. The authors report a comprehensive study examining the smoking independent pathologic changes in human nasal epithelial tissue and demonstrating the multimodal functional consequences of a5SNP in a transgenic mouse model with multiple environmental triggers. The authors also put forward a rational mechanism and pathways involved in the observed pathophysiology using cultured airway epithelial cells. Overall, the authors provide a robust study elucidating a causal pathway in the development of COPD, including a possible drug target and therapeutic option. While this study has several prominent strengths, we have several issues that we feel should be addressed prior to publication.

We would like to thank the Reviewer for this very positive overall evaluation of our study. In the following, we would like to address the issues raised.

1) Human association data:

a) The identification of the locus and causal variant (page 3) is confusing, e.g. "chromosome 15q25, a human haplotype encompassing three genes". The haplotype is not the same as the locus, and there are many more genes than the three mentioned. In fact, there is evidence about IREB2 as an effector gene at this locus, and also for more than one signal. There is no discussion of the extensive literature on fine mapping, identifying the causal variant, etc -- while much of this data supports investigation of the nonsynonymous SNP, the lack of mention is potentially misleading about the complexity of this locus.

We have now added a more detailed introduction to the locus. We would also like to point out that the group of one of the senior authors, UM, has been studying this same locus for their implication in smoking, schizophrenia, alcohol, and cocaine addiction, the work is published. This has been reviewed recently, Maskos *JNeurochem* 2020, and clearly points to the importance of the alpha5 coding SNP. It is the only amino acid change in the locus, and exerts a strong effect because of its crucial role in the function of the *brain* nicotinic receptor. Here, we demonstrate a clear functional difference in *airway epithelial cells*, as a mechanistic basis for its contribution to lung pathology. This does of course not exclude additional contributions from expression level changes of *CHRNA3* and *CHRNA4*. This is a point we discussed in our recent review for the *brain* receptors (*Neuropharmacology* 2020; 177:108234), and it will be addressed in future work for the present system.

We have now also added a discussion of the *IREB2* locus, based on the work of Cloonan et al (*Nat Med* 2016).

b) The idea that "solely 20-50% are tobacco smokers" is a concerning and inaccurate statement. Certainly one could identify a population where the smoking prevalence is very low, but studies that have identified this locus for COPD have not studied these populations, and from a public health standpoint this statement could very easily be misconstrued.

We completed our statement to avoid misconception. We wanted to highlight the fact that especially in developing countries a majority of patients are not tobacco smokers, but are exposed to other sources of smoke, like indoor fire places.

c) The human genetic evidence requires additional discussion. The largest and most recent study including the UK Biobank by Shrine et al does not identify any effect of this locus on lung function in non-smokers. This finding seems to argue that there may be issues in the relevance of their cell type selection and the relevance of the mouse model to human disease, and also raises the question whether issues in accurately ascertaining smoking history could have led to the identification of an association in 'non-smokers' or after adjusting for smoke exposure.

We have substantially increased the discussion of human genetic evidence, see pp 3-4 of the revised version. Yes indeed, establishing a smoking history is difficult in human subjects. This is also one of the main reasons why we wanted to focus on experiments not involving nicotine or smoke exposure, to be absolutely certain to create model systems for the effects in "non-smokers".

Our project was inspired by the study of Wilk, Shrine et al (*Am J Respir Crit Care Med* 186: 622–632), who identified the *CHRNA3/A5* region as a genetic risk factor for airflow obstruction, and concluded that "The region was also modestly associated among never smokers". Their "Gene expression studies confirmed the presence of *CHRNA5/3* in lung, airway smooth muscle, and bronchial epithelial cells." It was later confirmed in several genetic studies including

the recent work from Hopkins et al. (*Thorax* 2021; 76:272-280) where rs16969968 was found to **independently** confer risk of lung cancer, COPD and smoking intensity. We therefore feel to have targeted the correct cell types in our study. We added the only study where rs16969968 was not found associated with lung function as mentioned by the Reviewer. In the Shrine et al study from 2019 (*Nature Genetics* 51: 481-493), the "sentinel" SNP rs17486278 for *CHRNA5* was excluded at a p value of 3.964E-8, close to the cut-off. The authors state that "neither was significantly associated with lung function among never smokers and so both were excluded from further analysis", although no data are shown. Since raw data are not available to screen the millions of SNPs that were identified but not found statistically associated with lung function, we believe that the absence of rs16969968 may solely be due to the strong cut-off that was selected to reach significance ($p < 10^{-9}$).

d) The analysis of the human subject is problematic (page 5, paragraph 1, and others). The genetic data does not account for ancestry and possible population substructure - the allele frequency varies greatly by population. In addition, the association with cigarette smoking exposure makes controlling / adjusting for this covariate absolutely critical. Dividing up smokers into three somewhat arbitrary groups likely underestimates the contribution of smoking -- in fact, using their three groups of smokers, there appears to be no effect of smoking on epithelial inflammation, and smokers appear to have similar amounts of normal epithelium, which brings into question whether their smoking ascertainment or analysis is correct (Table 1 and Supplementary Table 1). Use of current vs former, duration, and packs per day (or pack-years) are standard approaches for adjustment and should be used.

We agree with the Reviewer that our smoking assessment was incomplete. The smoking habits are rarely evaluated in patients admitted for polypectomy, thus, we have only access to partial clinical information (status and date of withdrawal). In addition, the large majority of patients were non-smokers (n=123 vs smokers/ex-smokers n=26) so we could not establish statistics on the three classical groups (smokers, ex-smokers, and non-smokers).

To improve the manuscript, we restricted our analysis to non-smokers and modified our analysis to show that respiratory epithelial remodeling and inflammation is seen in α 5SNP patients in the absence of smoking.

Moreover, all our patients were Caucasians, where the allele frequency observed is on the order of 70% which is similar to the 58,3% reported from the 9270 non-hispanic whites from Hopkins et al. (*Thorax* 2021; 76:272-280).

The authors also performed a univariate screen to select covariates to include in a multivariable logistic regression to determine the effect of α 5SNP genotype on odds of inflammation. While univariate screen should pick up most relevant confounders, there may be interaction effects that may not appear without other variables. The authors have a large enough sample size to consider including the relevant available covariates without concern for overfitting, and there do not appear to be any obviously colinear variables.

We agree with the Reviewer and therefore included the relevant available covariates in the *Results*.

2) Cell types:

The authors use nasal polyps as a proxy for airway epithelium. basal cells..." It is understandable and justifiable why they chose nasal polyps, but there are substantial differences between different types of airway epithelium, and identification of the causal or effector cell types for the pathology remains unanswered. Similarly, (page 14, paragraph 3) they used tracheal basal cells -- "we used a model of local expression...including tracheal. Perhaps this is an experimental limitation of murine models, but it is unclear why the authors focused on tracheal pathology in assessing the role of α 5SNP. While there are hallmark tracheal changes sometimes seen in COPD (ie sabersheath trachea), the airway changes in COPD are typically seen more distally (<https://doi.org/10.1186/s12931-019-1017-y>). The authors should provide further rationale for the use of tracheal samples for analysis rather than more distal airway tissue.

Our choice is justified by the fact that mouse trachea or human polyps are both lined with respiratory epithelia, and represent relevant models to study the airway remodeling modifications, similar to those observed in COPD. In addition, we would like to point out that in a number of experiments, Figures 2b/c, 3a/b, Figures S3, S4, S6a, we identified phenotypes in **distal airways** and the **broncho-alveolar junction**. Altogether, our data show that the α 5SNP polymorphism was associated with alterations of different subtypes of airway epithelium present in both upper (including nasal epithelium) and distal airways.

3) Mouse data:

a) In some situations, the heterozygous data is presented, and in other cases, only the homozygous variant is presented. Is there a justification for excluding the hets if they are available? Similarly, their prior study used male mice, but in this study says "mice were... sex-matched"? More information on the sex balance would be helpful.

For studies analysing behaviour in our work addressing the *neural* consequences, we have indeed limited the dissection to males, a standard procedure in neuroscience. Here, both sexes were used, as indicated. Every time we had sufficient heterozygous animals for experimentation, they were included.

We would like to point out that when using our lentiviral vectors on an $\alpha 5$ KO background for the transduction *in vitro* or *in vivo*, no "heterozygous" expression can be achieved.

b) Given the substantial effects of this gene on behavioral phenotypes described in their previous work, is it possible that some of the effects they see are modified by behavior -- for example differences in socialization and effect on stress, diet and effects on nutrition / weight? Were these behavioral effects measured and accounted for?

We have carried out, and published, a more detailed analysis of behavioural phenotypes in transgenic rats expressing the coding $\alpha 5$ SNP. While reward processing is altered in SNP rats for nicotine, alcohol, cocaine, no differences were found in acquisition of food seeking, nor in weight. This was published in Besson et al, NPP 2019.

In addition, weight and general behaviour was systematically monitored during the experiments. A sentence was added in the *Methods* section. As an example, regarding the measurement of the respiratory function, there was no significant difference in weight (WT: 27 ± 1.4 g; HT: 29.01 ± 1.45 g; HO: 27.45 ± 1.66 g, n = 16-20)

c) Page 36, Figure 2: the would consider the analysis of the age effect different between the two groups via test for an interaction term (see: <https://www.nature.com/articles/nn.2886>).

Although this test would be useful to evaluate the age effect on a biological parameter considering two groups of different age, our aim here is only to demonstrate that the difference in MLI is seen in aged mice considering the two genotypes. Since it was confusing we separated the plots to avoid confusion.

4) COPD phenotypes:a) Page 3, paragraph 3: "Other than emphysema..." May be better stated as a subtype of COPD or in addition to emphysema.

This has been reformulated, p4.

b) Page 4, paragraph 1: "...COPD lesions..."

This has been rephrased, p5.

c) Page 6, paragraph 2: "...an increase in the inspiratory capacity..." this is not a hallmark of lung function changes in COPD; humans with COPD have decreased inspiratory capacity owing to increased functional residual capacity via increased residual volume

Indeed, there are some differences between respiratory function in mice and humans. Increased inspiratory capacity was associated with development of emphysema in mice. As underlined by functional and histopathological analyses, $\alpha 5$ SNP was associated with emphysema explaining the increased inspiratory capacity. We have further discussed this discrepancy in the revised version.

d) Page 10, paragraph 1: "...COPD-like lesions..." Probably best to define this using more commonly used terms – emphysema vs chronic bronchitis – or pathologic changes characteristic to COPD

This has been further detailed, p12.

e) Page 16, paragraph 3: "The cardiac pulmonary lobe..." Classically the upper lobes/zones are the most involved in COPD, which makes the choice of a "middle" lobe questionable. The authors nonetheless demonstrate pathologic changes, which may make the choice moot.

The description in the *Methods* was not clear. We used the left lobes for protein analysis and the remaining lobes for histological analysis. Therefore, upper and lower zones were both analysed. We modified the text in the *Methods*.

Additional comments:

a) 3) Page 8, paragraph 1: "Importantly, pre-treatment of mice with Etanercept..."

There is no mention of this experiment in the methods section. While demonstration of translation of findings to therapeutics is important, the authors should include discussion of rationale and transparency regarding choice of drug.

The description has been added to the *Methods* section. Etanercept is an inhibitor blocking the interaction of TNF α with its ligands, frequently used in dermatology and rheumatology, justifying our choice for our experiments.

In general, given the extensive number of experiments, a diagram or table of the experiments could be helpful. We have added a scheme in **Figure 1** to highlight the different experimental systems, human and mouse, and the different epithelia targeted, nasal, bronchial, alveolar.

b) Page 8, paragraph 1: How do the authors account for no increase in TNF- α in BALF of α 5SNP mice but increase in TNF- α in α 5KO mice following α 5SNP transduction?

We did not describe sufficiently the methods associated with this analysis.

The detection of cytokines was performed in two modalities: BALF for the distal component, and aspiration for the proximal component. The molecular content secreted by various cell types in different airway regions may explain the differences. During the BALF procedure, we diluted the secretions present within the airways in contrast to the aspiration procedure. This might also explain the lack of detection for TNF- α levels in the BALF.

We added the modality of collection in the *Methods*.

c) Page 7, paragraph 1: "All these findings..." This sentence is a conclusion and does not belong in the section reporting results

We removed this sentence.

d) Page 8, paragraph 2: "Thus, under conditions of airway epithelium aggression..."

Aggression does not seem to be the correct word to use in this case. Perhaps stimulation or agitation would fit better.

It has been replaced with "stimulation".

Page 11, Paragraph 1 -- what is meant by "alpha5SNP intervened"?

This has been rephrased.

e) Page 12, paragraph 2: "...a frequent transient prolonged squamous cell metaplasia." This phrase is unclear and transient prolonged is a bit of an oxymoron.

This has been rephrased.

f) Throughout: the use of HT and alpha5SNP as terms for the heterozygote and the homozygous models is confusing (see, for example, Supplement, Page 2).

This has been changed in the whole manuscript to HT α 5SNP, HO α 5SNP, and WT α 5.

g) I find it hard to interpret the video - there is no control group, and no description of what we are seeing.

The video was meant to illustrate the calcium imaging experiments. The differences between calcium transients in wild-type versus mutant epithelial cells are not detectable with the "naked" eye, and need image analysis. We have therefore removed the video.

h) Page 24, Paragraph 2 -- what were the reasons for dichotomization of the quantitative or ordinal scales?

To our knowledge and according to biostatistics, logistic regression models provided OR that better reflect the biological signification of each analysed parameter, and are classically found in the literature in order to compare data in various studies.

i) Page 35, Figure 1: 'Chorionic' is not a commonly used term outside of fetal anatomy, and is somewhat confusing -- is there an alternative that can be used?

This has been changed to "Lamina propria".

j) Page 36, Figure 2: Need to clarify the time period for experiments (are they 24 weeks, except as noted)?

We modified the figure legends to clarify the time period for all the panels.

k) In general, it would be much more informative to plot the individual data points, rather than box plots (particularly those with wide confidence intervals), e.g.

<https://journals.plos.org/plosbiology/article?id=10.1371/journal.pbio.1002128>

We formatted all the data to show dot plots.

l) The PV curves for the heterozygotes appear to have a higher compliance than the wild type and the alpha5SNP - can the authors explain this phenomena?

Static compliance (Cst) reflects the intrinsic elastic properties of the respiratory system including the chest wall. As reported in Figure 3 and the Supplementary Figures S8 and S9, the inflammatory reaction present in HT and HO α 5SNP mice are different between both groups. Namely, HT mice do not present a higher recruitment of iNKT and T γ δ cells and do not exhibit the production of CXCL1 and CCL5. Moreover, we have not measured the properties of lung muscles and structures potentially explaining the alteration in elasticity. Additional assessments are required in order to demonstrate this link with the increased CSt although this is outside the scope of the present study.

Reviewer #2 (Remarks to the Author):

This is an interesting study on the role of the alpha5 nAChR in COPD. Although well conducted, the study needs some more work.

It has been shown that similar to chromaffin cells in the adrenal gland airway epithelial cells express multiple subtypes of nAChRs (including the alpha5) that regulate the secretion of catecholamines which function as autocrine growth factors for airway epithelial cells and lung adenocarcinomas derived from them, a response mediated by binding of the catecholamines to adrenergic receptors that stimulate the activation of AC resulting in the formation of intracellular cAMP.. All nAChrs are cation channels which depolarize the cell membrane in response to the influx of cations. Even though in the present study the preferential influx of calcium through the channel of the alpha5nAChr was reduced by the SNP, membrane depolarization obviously still occurred likely via compensatory increase in the influx of non calcium cations as evidenced by the observed increase of AC. nAChRs do not directly increase AC but only obtain this response in an indirect fashion via the nAChR-induced release of catecholamines.

The authors need to additionally measure catecholamine release in their experimental systems in order to obtain a correct interpretation of their data.

We have not been able to find an extensive literature on the role of catecholamines in *lung* epithelial cells. However, there is a publication analysing the role of the nicotinic receptor subunits $\alpha 3$ and $\alpha 5$ in pancreatic cancer cell lines. We have therefore tried to measure the catecholamine levels in our cultures of cell sorted epithelial cells, as described here:

Murine tracheal epithelial cells (MTEC) were isolated from WT or a5SNP tracheas as described in the manuscript. After isolation, MTEC were resuspended in KFSM (Keratinocytes serum free medium, Gibco), expansion medium and expanded in T75 flasks until they reached 80%-90% of confluence as described in **(1)**.

MTEC were dissociated and plated at 2×10^4 for nicotine (Sigma) treated cells and 2.5×10^4 for cells treated with scratch. Cells were maintained in complete medium until they reached 65-70% of confluence for nicotine treated, or 100% for cells treated with a scratch. Then, they were switched to basal medium during 24h.

For the first part of the experiment, cells were untreated or incubated with 500nM, 1 μ M or 10 μ M of nicotine during 30 min, as described in **(2)**. The culture media, containing secreted catecholamines, were collected and to prevent catecholamine degradation, 1mM of EDTA (Sigma) and 4mM of sodium metabisulfite (Sigma) was added before freezing at -80°C.

For the second part of the experiment, in wells with 100% confluent cells, a lesion was carried out as a crossed scratch with a tip. Then, the culture media, containing secreted catecholamines, were collected 6h and 15h later, and to prevent catecholamine degradation 1mM of EDTA and 4mM of sodium metabisulfite was added before freezing at -80°C.

Quantitative analyses of secreted adrenaline, noradrenaline and dopamine of 4 samples per treatment group were conducted using the 3-Cat research ELISA kit (ImmuSmol) following the manufacturer's recommendations. Absorbance of samples was read using an iMark microplate reader, Biorad, at 450 nm primary wavelength with a 650 nm reference wavelength.

We were not able to properly analyse the results since the values obtained were not compatible with the analytical sensitivity of the kit.

Unfortunately, due to supply problems of culture medium, because of the current health crisis, we were not able to reproduce the complete experiment with a larger number of cells, but we were able to test the exposure to 10 μ M of nicotine and analyze 15 hours after lesion with 5 times more cells. But again, secreted catecholamine levels were not sufficient to reach the analytical sensitivity threshold of the test kit.

We conclude that in our experimental system potentially very small amounts of catecholamines are present, not detectable with the currently available kits. But it will of course be interesting to pursue this issue in future work, and establish a potential role of catecholamines in the physiology and pathophysiology of bronchial epithelial cells.

1. Eenjes, E., Mertens, T. C. J., Buscop-Van Kempen, M. J., Van Wijck, Y., Taube, C., Rottier, R. J., and Hiemstra, P. S. (2018) A novel method for expansion and differentiation of mouse tracheal epithelial cells in culture. *Sci. Rep.* **8**, 1–12
2. Al-Wadei, M. H., Al-Wadei, H. A. N., and Schuller, H. M. (2012) Pancreatic cancer cells and normal pancreatic duct epithelial cells express an autocrine catecholamine loop that is activated by nicotinic acetylcholine receptors alpha3, alpha5, and alpha7. *Mol. Cancer Res.* **10**, 239–249

Reviewer #3 (Remarks to the Author):

Polymorphisms in the CHRNA3 and CHRNA5 genes have been found and replicated many times in genome wide association studies for lung function in the general population. The general assumption has been that these polymorphisms relate to smoking addiction and some data in the literature support this contention, whereas other data seems to support this to be a smoking-independent risk factor. In fact, several studies have argued, that nicotinic receptors have additional roles in peripheral tissues (beyond neurotransmission) in governing inflammation, epithelial cell proliferation and differentiation and others. For the most part, these studies have addressed the role of the alpha7 nAChR in such responses. The role of the alpha5 nicotinic receptor has been studied mostly in the context of lung cancer development. The current work shows a major role for CHRNA5 and for polymorphisms of CHRNA5 specifically, in the regulation of epithelial cell biology in the nose and lung. This is a major finding, and in fact long-awaited data, that will have a major impact on the field. Nonetheless, I do have some concerns with the data presented and the data interpretation that I would like to discuss.

We would like to thank the Reviewer for this very positive overall evaluation of our study. In the following, we would like to address the issues raised.

1. I had difficulties connecting the nasal polyposis data in humans to the goblet cell hyperplasia data in the intrapulmonary airways of the alpha5SNP mice. In this respect, I recommend the following changes:

a. Please do not refer to nasal polyposis as a COPD-like condition (Figure 1 title) as it is not. It does provide rationale for a role of CHRNA5 in epithelial abnormalities and I suggest to define it as such.

This has been changed.

b. The paper would benefit enormously from data linking CHRNA5 genotype to epithelial cell characteristics of the intrapulmonary airways. This would make the work much more relevant to COPD.

We would like to point out that we analysed here main features of COPD: *respiratory epithelial remodeling, emphysema and inflammation* found both in KI mice expressing the SNP alpha5 channel protein and in well-characterised COPD lesions in humans. This led us to propose that the α 5SNP channel protein is involved in these lesions. The data presented in Figure 2b-c, Figure 3, Figures S4 to S9 were gathered on intra-pulmonary airways.

c. In the absence of such data, the context of the human data should be rephrased and more carefully positioned. Furthermore, nasal epithelial data in the mouse would be very helpful in that case to better link the human and the mouse data.

We rephrased the human data so that they better reflect our findings on COPD-like lesions in the mouse.

As suggested by the Reviewer, we analysed the murine nasal epithelia in two experimental modalities in order to better link the human and mouse data. In *untreated* 54 week old α 5WT and HO α 5SNP mice, we performed a histological analysis on nasal respiratory epithelia. As we reported in human nasal polyps, we found a significant increase of secretory cells in HO mice (see new Figure S2b). Nonetheless, we could not evaluate the inflammation on the extremely reduced sub-epithelial lamina propria. Therefore, we thought to induce inflammation with cumene hydroperoxide instillation. Not even under these conditions could we observe an inflammation of the chorion. We conclude that the murine nasal cavity is an adequate model to evaluate epithelial remodeling, but not the inflammation.

d. The polidocanol data is interesting; however I disagree with the assertion that this is a COPD model. It is an upper airway injury model and valuable in its own right, but does not model COPD.

We agree with the Reviewer and we rephrased.

e. If human COPD data cannot be obtained, however, at least in the opinion of this reviewer, the title needs to be adapted to reflect that the work is on the role of alpha5 nicotinic receptor on nasal and airway epithelial remodeling and not on COPD. There is no COPD data in the paper.

We agree with Reviewer and changed the title to "...COPD-like lesions".

2. Presumably, HT means heterozygous but I could not confirm this in the paper.

It was only mentioned in the *Introduction*, but for clarity, we added it also in the *Methods* section.

3. The authors have the alpha5KO mice available but I could not find any data in the paper on the phenotype of these mice in comparison to WT mice. Is endogenous expression of alpha5 sufficient for the epithelial cell changes observed, or is overexpression of the channel protein (either local in epithelial cells or whole body) required to expose the effects? In my opinion, the KO mouse could be valuable in assessing this important question, for example by using a cigarette smoke exposure model which would directly link nicotine exposure to pulmonary changes as a function of the absence or presence of the alpha5 receptor.

The alpha5 KO mice have been used for several experiments presented. In Figure 4, lentiviral vectors are used to express the $\alpha 5$ WT subunit, the $\alpha 5$ SNP, and eGFP on an alpha5 KO background. In Figure 4c, d, e, f, the "GFP" label thus refers to the $\alpha 5$ KO. Similarly, in Figure 5a, b.

The suggestion of the reviewer to use cigarette smoke exposure in the KO mouse is clearly very interesting, but our present study was intended to address the role of the $\alpha 5$ SNP in the airways and lungs in the absence of nicotine and cigarette smoke exposure. We will explore their role in a further study in $\alpha 5$ SNP mice

4. It is unclear to me what WT means in Figure 2/3. Are these wild-type mice or mice overexpressing the WT alpha5 channel protein? If they are wild-type mice, how can the authors deduce receptor overexpression effects from SNP-specific expression effects?

The "WT" used in the experiments for Figures 2 and 3 are wild-type C57BL6/J littermates of the rs16969968 SNP knock-in mice. In the wild-type mice and SNP knock-ins, there is no over-expression of the subunits.

5. There is one major issue I cannot get my head around. The minor A/A allele produces a less efficacious channel than the WT allele. Yet, in the present study, only expression of the minor allele produced functional effects in the alpha5 KO mice. Mice expressing the more effective alpha5 WT allele were unaffected in comparison to alpha5 KO mice expressing GFP. This does not make sense to me: how can absence of expression and overexpression of the WT receptor have similar effects with overexpression of a less active receptor having differential effects? This discrepancy needs to be adequately explained and supported by data.

We would like to point out that in functional assays, like Figure 5a, there is a clear difference between the expression of the WT channel, and the complete $\alpha 5$ KO. Expression of the $\alpha 5$ SNP channel results in an intermediate phenotype, so the calcium permeability is $\alpha 5$ WT > $\alpha 5$ SNP > $\alpha 5$ KO.

Also in the experiments presented in Figure 4, for example 4c, there is a net difference between the WT and $\alpha 5$ KO in the lesion study. Similarly, Figure 4f shows that cell proliferation is also increased in both $\alpha 5$ KO mice transfected with GFP and SNP LV vectors compared to $\alpha 5$ KO mice transfected with WT LV vectors. Thus, in our opinion, there is not a real discrepancy between *in vivo* and *in vitro* studies comparing the behaviour of transduced mouse tracheal cells.

REVIEWER COMMENTS

Reviewer #1 (Remarks to the Author):

The authors have addressed most of the comments adequately, and the revised manuscript is much improved. My remaining comments:

Major:

1. The statements about COPD risk in non-smokers are slightly improved, but still of concern. It is not clear where the 20-50% comes from (having examined the references) -- it should be noted that it is a mistake of assuming the attributable risk adds up to 100%. The main data source is reference 16, but if one actually attempts to calculate the percentage of GOLD2+ COPD subjects based on the percentages shown in fact this number is often >90% depending on the site and sex (see, for example, the Lexington, KY US men).

I would favor leaving the sentence about "substantial percentage", and the citations shown, without getting into potentially misleading numbers.

2. The review of the prior GWAS literature is still problematic.

a. Most studies control for smoking as a covariate. This has been done by Shrine et al and Sakornsakolpat et al in the most recent GWAS of lung function and COPD. Both the Shrine and Sakornsakolpat analyses identified the 15q25 regions, at either genome-wide or close to genome-wide significance. However, both of these analyses, relying heavily on UK Biobank data, did NOT find an association in never smokers.

3. The statement, "Interestingly, two thirds of patients were HT or HO α 5SNP, pointing towards a potential strong association of this polymorphism with clinical outcome (Table S1)." should be removed.

It is unclear what the authors are trying to imply here, but it appears as if they are trying to make an argument for genetic association based simply on the allele frequency in their entire cohort without a control or any statistical analysis.

Minor:

1. The abstract states, "Chronic Obstructive Pulmonary Disease (COPD) is a major cause of morbidity and mortality, generally smoking-linked." It is unclear, whether "smoking-linked" is a modifier of COPD, morbidity, mortality, or all three. Suggest, COPD is a generally smoking-linked major cause...? Or something to clarify.

2. Note that the use of mediation analysis to examine the association between the 15q25 region and the outcome of interest has been performed by Siedlinski et al (PMID: 23299987), and also is consistent with and supports the work of Wilk et al and recent work of Hopkins et al. Typo in this section "SPN" should be "SNP".

3. To clarify the comment,

1. The abstract states, "Chronic Obstructive Pulmonary Disease (COPD) is a major cause of morbidity and mortality, generally smoking-linked." It is unclear, whether "smoking-linked" is a modifier of COPD, morbidity, mortality, or all three. Suggest, COPD is a generally smoking-linked major cause...? Or something to clarify.

2. Note that the use of mediation analysis to examine the association between the 15q25 region and the outcome of interest has been performed by Siedlinski et al (PMID: 23299987), and also is consistent with and supports the work of Wilk et al and recent work of Hopkins et al. Typo in this section "SPN" should be "SNP".

3. Regarding minor comment h) "what were the reasons for dichotomization of the quantitative or ordinal scales?" the authors do not appear to have understood the question. The question refers to the issues and concerns with dichotomizing inherently continuous data: PMC1458573, including increasing the possibility of a false positive result.

Reviewer #2 (Remarks to the Author):

COPD is a disease of the lung periphery involving bronchioles (synonyms: small airways, peripheral airways) and alveoles, resulting in emphysema. The murine tracheal epithelial cells used by the authors to conduct additional reviewer-requested studies are incorrect as they are comprised of a reparatory epithelium of ciliated cells mucous producing goblet cells and basal cells (which have stem cell function). By contrast, small airway epithelium is comprised of ciliated cells and non-ciliated secretory cells (also referred to as Clara cells) , with the latter having stem cell function. The authors should have used primary small airway epithelial cells from normal human lung and COPD human lung instead both of which are commercially available from the American Type Culture Collection (ATCC).

Reviewer #3 (Remarks to the Author):

I thank the authors for answering my questions, which was done very satisfactorily. I keep on having difficulties with the framing of parts of the data as COPD or COPD-like as it really is not (particularly the nasal polyposis). I have some well-meant suggestions below. I wish to emphasize that it is my every intention to ensure that the respiratory community accepts the conclusions from the paper.

I think it should be perfectly possible to let COPD be the context of the paper without stressing too much that all the models are very COPD-like. This goes for several spots in the manuscript, and my advice is that the authors carefully go over each of these. I provide 2 examples:

The last paragraph on page 4:

We have used mouse models, particularly "humanized" mice expressing $\alpha 5$ SNP31, to determine its role in the development of a COPD-like phenotype without any experimental manipulation

why not simply state ...development of a lung phenotype....or development of lung pathology or epithelial remodeling and inflammation

The first sentence of the materials and methods:

The objective of this work was to study the role of $\alpha 5$ SNP in the main features of COPD: airway epithelial remodeling, emphysema, and inflammation.

This sentence could also be written as:

The objective of this work was to study the role of COPD-associated $\alpha 5$ SNP in airway epithelial remodeling, emphysema, and inflammation.

These are just examples and of I leave it up to the authors discretion to decide, but in my view the authors over-emphasize the relevance of their models for human COPD, which I think is unnecessarily distractive and risky. The context of the work is COPD and is clear; the models recapitulate part of the processes but not its entirety and that is also clear. I do not think it is needed or helpful to stress COPD or COPD-like that much.

REVIEWER COMMENTS

Reviewer #1 (Remarks to the Author):

The authors have addressed most of the comments adequately, and the revised manuscript is much improved. My remaining comments:

Major:

1. The statements about COPD risk in non-smokers are slightly improved, but still of concern. It is not clear where the 20-50% comes from (having examined the references) -- it should be noted that it is a mistake of assuming the attributable risk adds up to 100%. The main data source is reference 16, but if one actually attempts to calculate the percentage of GOLD2+ COPD subjects based on the percentages shown in fact this number is often >90% depending on the site and sex (see, for example, the Lexington, KY US men).

I would favor leaving the sentence about "substantial percentage", and the citations shown, without getting into potentially misleading numbers.

As suggested by Reviewer #1, we removed the discussion concerning the percentage to avoid wrong assumptions.

2. The review of the prior GWAS literature is still problematic.

a. Most studies control for smoking as a covariate. This has been done by Shrine et al and Sakornsakolpat et al in the most recent GWAS of lung function and COPD. Both the Shrine and Sakornsakolpat analyses identified the 15q25 regions, at either genome-wide or close to genome-wide significance. However, both of these analyses, relying heavily on UK Biobank data, did NOT find an association in never smokers.

Following the comments of the Reviewer, we have removed the reference.

3. The statement, "Interestingly, two thirds of patients were HT or HO α 5SNP, pointing towards a potential strong association of this polymorphism with clinical outcome (Table S1)." should be removed.

It is unclear what the authors are trying to imply here, but it appears as if they are trying to make an argument for genetic association based simply on the allele frequency in their entire cohort without a control or any statistical analysis.

As suggested by Reviewer #1, we removed the statement concerning this conjecture.

Minor:

1. The abstract states, "Chronic Obstructive Pulmonary Disease (COPD) is a major cause of morbidity and mortality, generally smoking-linked." It is unclear, whether "smoking-linked" is a modifier of COPD, morbidity, mortality, or all three. Suggest, COPD is a generally smoking-linked major cause...? Or something to clarify.

We thank Reviewer #1 for his help in clarifying this sentence which has been changed to: "Chronic Obstructive Pulmonary Disease (COPD) is a generally smoking-linked major cause of morbidity and mortality".

2. Note that the use of mediation analysis to examine the association between the 15q25 region and the outcome of interest has been performed by Siedlinski et al (PMID: 23299987), and also is consistent with and supports the work of Wilk et al and recent work of Hopkins et al. Typo in this section "SPN" should be "SNP".

We thank the Reviewer and have added the Siedlinski reference to the MS. They state in their summary of results with 3,424 COPD cases and 1,872 unaffected controls that "analysis revealed that effects of two linked variants (rs1051730 and rs8034191) in the *AGPHD1/CHRNA3* cluster on COPD development are significantly, yet not entirely, mediated by the smoking related phenotypes. Approximately 30% of the total effect of variants in the *AGPHD1/CHRNA3* cluster on COPD development was mediated by pack years. Simultaneous

analysis of modestly ($r^2 = 0.21$) linked markers in *CHRNA3* and *IREB2* revealed that an even larger (~42 %) proportion of the total effect of the *CHRNA3* locus on COPD was mediated by pack years after adjustment for an *IREB2* single nucleotide polymorphism." This would suggest that potentially 60% of the effect is not linked to smoking.

3. Regarding minor comment h) "what were the reasons for dichotomization of the quantitative or ordinal scales?" the authors do not appear to have understood the question. The question refers to the issues and concerns with dichotomizing inherently continuous data: PMC1458573, including increasing the possibility of a false positive result.

We agree with the Reviewer and are aware that dichotomising continuous variables may lead to false-positive results to some extent. Here, linear regression has also been used during statistical analysis. It leads to the same results for the variable "normal epithelium" (see table below). But the model was not globally significant for the "secretory cell hyperplasia" variable. Although it was included in the remodelling vs non-remodelling analysis (and as such in the "normal epithelium"), we thought that this component brought additional information that specified the type of remodelling that we observed. The fact that it was only statistically significant with linear regressions prompted us to favour the model that we presented. Regarding the "inflammation" variable, we thought that it made more biological sense to regroup the 4 initial categories into 2 categories with "low" and "high" inflammation (see figure below with detailed representation of values). Altogether, we chose to present dichotomised variables to help the reader with data visualisation.

Multivariate analysis of the % of normal epithelium with linear regression model in 123 human nasal polyps

	Normal epithelium Coefficient [95% CI]	p-value
Genotype		
WT	ref	
HT	-12.52 [-22.78 ; -2.26]	0.017*
HO	-18.09 [-31.14 ; -5.03]	0.007*
Age	0.07 [-0.22 ; 0.36]	0.643
Gender		
M	ref	
F	-3.04 [-12.16 – 6.08]	0.511
Allergy/asthma		
No	ref	
Yes	4.51 [-4.73 ; 13.75]	0.336

Inflammation detailed values distribution for 123 human nasal polyps according to the score defined in Figure 1b:

Reviewer #2 (Remarks to the Author):

COPD is a disease of the lung periphery involving bronchioles (synonyms: small airways, peripheral airways) and alveoles, resulting in emphysema. The murine tracheal epithelial cells used by the authors to conduct additional reviewer-requested studies are incorrect as they are comprised of a reparatory epithelium of ciliated cells mucous producing goblet cells and basal cells (which have stem cell function). By contrast, small airway epithelium is comprised of ciliated cells and non-ciliated secretory cells (also referred to as Clara cells) , with the latter having stem cell function. The authors should have used primary small airway epithelial cells from normal human lung and COPD human lung instead both of which are commercially available from the American Type Culture Collection (ATCC).

We agree with Reviewer #2 that COPD involves bronchiolar and alveolar remodeling, but also bronchial remodeling. Whether the origin of the disease finds its roots at the distal or proximal end of the lung is subject to debates, and both hypotheses are heavily discussed in the literature, including work from us and collaborators¹⁻¹¹. Considering the large heterogeneity in COPD, it is most likely that different patterns of pathogenesis and progression (co)exist as recently evidenced¹². Ultimately, COPD involves the remodeling of bronchial (respiratory epithelium), bronchiolar, and alveolar epithelia. This is the reason why we evaluated the three epithelia and their remodeling in our study. We identified $\alpha 5$ SNP-associated alterations in the three epithelia. In brief, with the associated figure panels and analysis (in mice except when mentioned in brackets):

Respiratory epithelium: Fig1 (human), Fig2a, Fig4a-g, Fig4h (human), Fig5, FigS1, FigS2, FigS10, FigS11, FigS12, FigS13

Bronchiolar epithelium: Fig2b, Fig3b, FigS3, FigS7

Alveolar epithelium: Fig2c, Fig3a, FigS4, FigS6

Concerning the additional experiment that we provided in the first round of revision to address the question of the release of catecholamines in our experimental system. We would like to point out that our experimental system for the "mechanistic" dissection is FACS-sorted (tracheal) epithelial cells harbouring $\alpha 5$ WT or $\alpha 5$ SNP (see Figures 4 and 5, and associated additional Figures). Therefore we measured catecholamine release in csBEC to provide complementary data as requested by Reviewer #2 in her/his original review.

Reviewer #2 suggests now to reiterate the measurement of catecholamine release on commercially available non-COPD and COPD small airway epithelial cells. Reviewer #2 may refer to human small airway epithelial cells (HSAEC), ATCC® PCS-301-010, and COPD HSAEC, ATCC® PCS-301-013. Since the genotype (including our SNP of interest) will not be provided, it is basically impossible to obtain cells from patients harboring homozygous (A/A) rs16969968 (12% of the population¹³) considering that the clinical characteristics of the commercial cells are "batch-specific" as mentioned on the company site. In this context, we will **not be able to determine the link between rs16969968 SNP and catecholamine** in COPD and non-COPD HSAEC. Additionally, the whole design of our present investigation constitutes the prelude for further studies on COPD biological material (including humans) as we mentioned in the conclusion of our manuscript. We believe that the experiment suggested by Reviewer #2 would ideally complement the next step of our project in COPD patients.

Nonetheless, considering that evaluating catecholamine release may ultimately help to refine the mechanism of action of $\alpha 5$ SNP, we propose here an experimental alternative. We measured catecholamine release in bronchoalveolar lavages (BAL) from our WT and HO mice to complement the potential molecular mechanism associated with rs16969968-induced

remodeling and inflammation. The BAL contain *bronchial*, *bronchiolar*, and *alveolar* secretions, therefore it is the optimal biological material to evaluate the potential differential release of catecholamines from epithelial cells in association with α 5SNP.

Analysis of secreted catecholamines in BALF:

We first carried out an experiment to quantify catecholamines in broncho-alveolar lavage fluid of old (42 to 47 week-old) and young (21 to 26 week-old) WT and HO α 5SNP mice without any stimulation. The idea was to use conditions close to the observed age-dependent pathology in **Figure 2c**.

Baselines of the concentration of adrenaline, noradrenaline and dopamine present in BALF of 4 animals per group were determined using the 3-Cat research ELISA kit (ImmuSmol) following the manufacturer's recommendations. Absorbance of samples was read using an iMark microplate reader, Biorad, at 450 nm primary wavelength with a 650 nm reference wavelength. Analytical sensitivity: Adrenaline 10pg/ml, noradrenaline 4pg/ml and dopamine 10pg/ml.

	Adrenaline pg/ml	Noradrenaline pg/ml	Dopamine pg/ml
WT old 1	109,85	ND	ND
WT old 2	ND	ND	27,92
WT old 3	47,23	75,25	35,31
WT old 4	ND	75,86	ND
WT young 1	ND	68,16	ND
WT young 2	ND	ND	ND
WT young 3	14,09	75,48	4,55
WT young 4	ND	71,86	ND
HO old 1	ND	76,69	11,71
HO old 2	15,93	73,04	ND
HO old 3	97,64	79,10	35,31
HO old 4	54,84	72,92	ND
HO young 1	ND	65,61	ND
HO young 2	ND	ND	ND
HO young 3	ND	76,42	2,36
HO young 4	ND	75,46	ND

ND : Not detectable

The levels of the different catecholamines analysed are close to or below the analytical sensitivity of the kit and it was not possible to detect them in many animals, particularly in young animals.

As in our study we demonstrate that COPD hallmarks were significantly amplified in mice expressing the mutated $\alpha 5$ subunit after exposure to an oxidative stress, we also carried out an experiment to quantify the catecholamine concentration in BALF after an exposure to cumene hydroperoxide, as described in the manuscript, where the results are presented in **Figure 3**. Fourteen to 20 week-old WT mice or HO for $\alpha 5$ SNP were intranasally instilled with 50 μ l of 3 mg/ml cumene hydroperoxide. Catecholamine analysis from BALF of 6 mice per group was performed at 1 or 3 days post-instillation.

Day 1:

Grey spots = WT mice ; Blue spots = HO $\alpha 5$ SNP mice

Day 1	Adrenaline pg/ml	Noradrenaline pg/ml	Dopamine pg/ml
WT	35,04	15,25	ND
WT	23,27	13,05	ND
WT	21,00	45,68	ND
WT	2,95	52,03	2,01
WT	22,57	47,87	16,92
WT	34,00	55,03	ND
HO	4,51	46,90	ND
HO	11,40	33,40	ND
HO	22,92	4,79	ND
HO	2,25	0,00	ND
HO	19,62	16,60	7,78
HO	8,08	26,88	ND

ND : Not detectable

At Day 1, no significant difference is observed between WT and HO mice. Even after stimulation, the level of dopamine stays below the analytical sensitivity of the kit, and we were able to detect it only in a few animals.

Day 3:

Day 3	Adrenaline pg/ml	Noradrenaline pg/ml	Dopamine pg/ml
WT	2,21	21,03	7,75
WT	ND	24,08	ND
WT	7,75	63,49	8,84
WT	38,92	91,04	11,21
WT	1,52	36,96	11,89
WT	ND	37,86	ND
HO	2,44	13,97	ND
HO	ND	49,19	ND
HO	19,35	11,16	7,16
HO	0,38	16,50	ND
HO	12,91	11,45	ND
HO	ND	18,34	ND

ND : Not detectable

At Day 3, no significant difference is observed between WT and HO mice for adrenaline and dopamine, whereas the noradrenaline levels are slightly lower in α 5SNP mice compared with WT mice but the concentrations detected remain extremely low. Statistical analysis: The normality of the distribution of the data was tested by a Shapiro Wilk test. Then depending of the normality of the data a t-Test or a Mann Whitney test was applied. (Noradrenaline D3, $p=0.026$)

We cannot justify to pursue this experimentation further in respect of the 3R in animal experimentation.

According to our results, we can conclude that catecholamines are barely detectable even in severely challenged mice. They likely do not play a major role in cellular mechanisms involved during airway remodeling in the experimental systems we use in our study. The original hypothesis of Reviewer #2 in the first round of reviewing was that the SNP would **increase** catecholamine levels, thus leading to enhanced AC activity. Although very tentative, our result on noradrenaline levels at Day 3 after lesion would indicate the opposite.

References

1. Pain, M. *et al.* Tissue remodelling in chronic bronchial diseases: from the epithelial to mesenchymal phenotype. *European Respiratory Review* **23**, 118–130 (2014).
2. Ganesan, S. & Sajjan, U. S. Repair and remodeling of airway epithelium after injury in chronic obstructive pulmonary disease. *Curr Respir Care Rep* **2**, 145–154 (2013).
3. Puchelle, E., Zahm, J.-M., Tournier, J.-M. & Coraux, C. Airway Epithelial Repair, Regeneration, and Remodeling after Injury in Chronic Obstructive Pulmonary Disease. *Proceedings of the American Thoracic Society* **3**, 726–733 (2006).
4. Hadzic, S. *et al.* Lung epithelium damage in COPD – An unstoppable pathological event? *Cellular Signalling* **68**, 109540 (2020).
5. Higham, A., Quinn, A. M., Caçado, J. E. D. & Singh, D. The pathology of small airways disease in COPD: historical aspects and future directions. *Respiratory Research* **20**, 49 (2019).
6. Gohy, S. *et al.* Altered generation of ciliated cells in chronic obstructive pulmonary disease. *Sci Rep* **9**, 17963 (2019).
7. Koo, H.-K. *et al.* Small airways disease in mild and moderate chronic obstructive pulmonary disease: a cross-sectional study. *The Lancet Respiratory Medicine* **6**, 591–602 (2018).
8. Sohal, S. S. Epithelial and endothelial cell plasticity in chronic obstructive pulmonary disease (COPD). *Respiratory Investigation* **55**, 104–113 (2017).
9. Crystal, R. G. Airway basal cells. The ‘smoking gun’ of chronic obstructive pulmonary disease. *Am. J. Respir. Crit. Care Med.* **190**, 1355–1362 (2014).
10. Shaykhiev, R. & Crystal, R. G. Early Events in the Pathogenesis of Chronic Obstructive Pulmonary Disease. Smoking-induced Reprogramming of Airway Epithelial Basal Progenitor Cells. *Annals ATS* **11**, S252–S258 (2014).
11. Perotin, J.-M. *et al.* Alteration of primary cilia in COPD. *Eur. Respir. J.* **52**, (2018).
12. Young, A. L. *et al.* Disease Progression Modeling in Chronic Obstructive Pulmonary Disease. *Am J Respir Crit Care Med* **201**, 294–302 (2020).
13. Hopkins, R. J. *et al.* Chr15q25 genetic variant (rs16969968) independently confers risk of lung cancer, COPD and smoking intensity in a prospective study of high-risk smokers. *Thorax* **76**, 272–280 (2021).

Reviewer #3 (Remarks to the Author):

I thank the authors for answering my questions, which was done very satisfactorily. I keep on having difficulties with the framing of parts of the data as COPD or COPD-like as it really is not (particularly the nasal polyposis). I have some well-meant suggestions below. I wish to emphasize that it is my every intention to ensure that the respiratory community accepts the conclusions from the paper.

I think it should be perfectly possible to let COPD be the context of the paper without stressing too much that all the models are very COPD-like. This goes for several spots in the manuscript, and my advice is that the authors carefully go over each of these. I provide 2 examples:

The last paragraph on page 4:

We have used mouse models, particularly "humanized" mice expressing α 5SNP31, to determine its role in the development of a COPD-like phenotype without any experimental manipulation

why not simply state ...development of a lung phenotype....or development of lung pathology or epithelial remodeling and inflammation

The first sentence of the materials and methods:

The objective of this work was to study the role of α 5SNP in the main features of COPD: airway epithelial remodeling, emphysema, and inflammation.

This sentence could also be written as:

The objective of this work was to study the role of COPD-associated α 5SNP in airway epithelial remodeling, emphysema, and inflammation.

These are just examples and of I leave it up to the authors discretion to decide, but in my view the authors over-emphasize the relevance of their models for human COPD, which I think is unnecessarily distractive and risky. The context of the work is COPD and is clear; the models recapitulate part of the processes but not its entirety and that is also clear. I do not think it is needed or helpful to stress COPD or COPD-like that much.

We agree with Reviewer #3 and we made the following changes to tone down the relevance to COPD in 8 sentences:

- Page 2: "They were significantly amplified after exposure to porcine pancreatic elastase, an emphysema model, and to oxidative stress with a polymorphism-dependent alteration of lung function"
- Page 4 last paragraph: "We have used mouse models, particularly "humanized" mice expressing α 5SNP31, to determine its role in the development of epithelial remodeling and inflammation without any experimental manipulation, and to established nicotine-independent murine models mimicking these alterations, after intra-tracheal porcine pancreatic elastase (PPE) treatment³² and oxidative stress"
- Page 15 first paragraph: "The objective of this work was to study the role of COPD-associated α 5SNP in airway epithelial remodeling, emphysema, and inflammation"
- Page 12 first paragraph: "In this study, we demonstrate a direct association and a nicotine- and smoke-exposure independent role of α 5SNP in the development of pulmonary lesions, beyond its implication in the increase of nicotine intake^{10,11}. We focused here on airway epithelial remodeling, emphysema, inflammation, impact of oxidative stress and wound repair.
- Page 12 second paragraph: "This propensity was confirmed in young α 5SNP mice by using a model of elastase-induced emphysema. Interestingly, these lesions were also

amplified by oxidative stress, provoked here by lipid peroxidation with cumene hydroperoxide mimicking the effects of pollutants”

- Page 14, last paragraph: “In conclusion, α 5SNP expression, both in heterozygous and homozygous forms, directly facilitates the development of COPD-like lesions, including airway epithelial remodeling and emphysema, by promoting lung inflammation, jointly acting with oxidative stress”

REVIEWERS' COMMENTS

Reviewer #1 (Remarks to the Author):

I appreciate the authors' responses.

I believe their presentation of the background and genetic evidence are now more accurate. A minor point, instead of removing the references to the Shrine et al and Sakornsakolpat et al studies, it may be more balanced to include the statement that there was no association in non-smokers, illustrating that there is conflicting evidence in non-smokers, however I would leave this to the authors' discretion.

The language updating their findings to be more specific and accurate with respect to murine COPD models versus human is improved. I note that the figures / tables in the rebuttal Day 3 appear to have an error (dopamine and adrenaline plots are swapped), and there are a large number of missing values.

Reviewer #2 (Remarks to the Author):

Additional experiments conducted have satisfied this reviewer's concerns.

Reviewer #1 (Remarks to the Author):

I appreciate the authors' responses.

I believe their presentation of the background and genetic evidence are now more accurate. A minor point, instead of removing the references to the Shrine et al and Sakornsakolpat et al studies, it may be more balanced to include the statement that there was no association in non-smokers, illustrating that there is conflicting evidence in non-smokers, however I would leave this to the authors' discretion.

We would like to thank the reviewer. We have added the statement and the two references.

The language updating their findings to be more specific and accurate with respect to murine COPD models versus human is improved. I note that the figures / tables in the rebuttal Day 3 appear to have an error (dopamine and adrenaline plots are swapped), and there are a large number of missing values.

We apologise for this oversight. We have corrected the Table. Please note that we only add values to the figures if they are within the sensitivity range of the kit. When outside of this range, they are indicated as ND, not detectable.